# Strategic Hypothesis Testing

**Yatong Chen**[*]

Max Planck Institute for Intelligent Systems

Tübingen AI Center

`yatong.chen@tuebingen.mpg.de`

**Safwan Hossain**[*]

Harvard University

`shossain@g.harvard.edu`

**Yiling Chen**

Harvard University

`yiling@seas.harvard.edu`

## Abstract

We examine hypothesis testing within a principal-agent framework, where a strategic agent, holding private beliefs about the effectiveness of a product, submits data to a principal who decides on approval. The principal employs a hypothesis testing rule, aiming to pick a p-value threshold that balances false positives and false negatives while anticipating the agent's incentive to maximize expected profitability. Building on prior work, we develop a game-theoretic model that captures how the agent's participation and reporting behavior respond to the principal's statistical decision rule. Despite the complexity of the interaction, we show that the principal's errors exhibit clear monotonic behavior when segmented by an efficiently computable critical p-value threshold, leading to an interpretable characterization of their optimal p-value threshold. We empirically validate our model and these insights using publicly available data on drug approvals. Overall, our work offers a comprehensive perspective on strategic interactions within the hypothesis testing framework, providing technical and regulatory insights.

## 1 Introduction

In data-driven decision making, the outcome assigned to an agent often depends on data submitted by them, creating the potential for misaligned incentives. The role of strategic behavior in decision making systems has thus become a burgeoning area of research in the machine learning community, from classification [1, 2], to regression [3, 4, 5] and beyond [6]. However, limited literature exists on this perspective for a widely used and influential process in regulatory and scientific settings: hypothesis testing. Widely used in clinical trials, scientific research, and technological innovations [7, 8, 9, 10], hypothesis testing serves as a foundational method for assessing whether the evidence provided by a participating agent is both statistically significant and convincing. Consider, for instance, the U.S. Food and Drug Administration (FDA), which oversees drug approvals by setting a p-value threshold $\alpha$ for submitted clinical trials. Pharmaceutical firms incur substantial costs by participating in drug development and running trials in the hopes of being approved. Although falsifying results has high reputational and legal risks, firms are free to decide *if* to participate and *how large* to run a trial. We argue these decisions are intimately shaped by the FDA's p-value threshold and the cost-benefit calculus of the overall process. We thus propose a principal-agent model to understand these nuanced strategic decisions surrounding participation and evidence collection for a hypothesis test.

---

[*]Equal contribution. Correspondence to `yatong.chen@tuebingen.mpg.de` or `shossain@g.harvard.edu`.

39th Conference on Neural Information Processing Systems (NeurIPS 2025).

Bates et al. [11] initiated this literature by casting how agents respond to regulatory approval thresholds as a Stackelberg game. We extend this framework to capture additional real-world complexities that are abstracted away in their model. Specifically, we extend the analysis beyond binary notions of effectiveness and fixed trial costs, allowing agents to strategically choose their trial size based on the expected benefit relative to marginal per-sample cost. This extension not only enriches the analysis of agent incentives but also provides a more nuanced understanding of how regulatory policies can influence participation and decision outcomes. This richer model, however, leads to a more complex relationship between the regulator-specified p-value and the resulting error rates that they wish to control – for instance, if marginal costs increase faster than the expected revenue gains, raising $\alpha$ could paradoxically reduce the likelihood of passing the hypothesis test. This is further complicated by the principal wishing to control any combination of Type I (false positive) and Type II (false negative) errors within our model. Indeed, while the broader literature focuses on only the former [11, 12, 13, 14], we argue that minimizing the rejection or non-participation of effective products is also consequential to decision-makers, especially when this is influenced by low revenue relative to costs, as is the case for low-cost and orphan drugs [15]. We outline our key contributions toward addressing these challenges below:

1. **Game-Theoretic Framework:** Like Bates et al. [11], we consider a principal-agent (Stackelberg game) framework but relax several key assumptions to adhere more closely to real-world settings. We allow agents to strategically choose their sample sizes as part of their best responses and account for the marginal cost of such samples. Further, we model the principal as aiming to minimize any combination of Type I and II errors.

2. **Analysis of Component-Wise Losses:** We rigorously analyze the principal's loss components as functions of the p-value threshold under agent best-responding behavior. Despite the complex interplay between several factors, the monotonicity of Type I and Type II errors as a function of the p-value threshold $\alpha$ is preserved in a *piecewise manner*. Specifically, there exists a critical threshold $\hat{\alpha}$ such that the error dynamics are monotonic within each segmented region: one for $\alpha < \hat{\alpha}$ and one for $\alpha \geq \hat{\alpha}$.

3. **Empirical Validation and Policy Implications:** We validate our theoretical findings through an empirical analysis of drug approvals for three major drug categories and show that under our model, the commonly used p-value of $0.05$ aligns well with median revenue thresholds in these markets. The results suggest our model captures the economic dynamics of this process and suggests new policy insights.

**Additional Related Works:** Our work is closely related to the literature on economic aspects of statistical testing, p-hacking, contract theory, and FDA regulatory policies. A growing body of research examines the economic incentives in statistical decision-making [12, 14, 16, 13, 17]. Shi et al. [13] extends the work of Bates et al. [11] (which is discussed above) by studying strategic hypothesis testing under general concave utility functions, providing bounds on the Bayes False Discovery Rate (FDR). Relatedly, Bates et al. [14] models the agent-principal interaction through a contract-theoretic lens for incentive alignment, differing from our Stackelberg game approach. Also related is the well-known issue of *p-hacking*, where researchers manipulate sample sizes or selectively report findings to artificially achieve statistical significance [18, 9, 10, 19, 8]. Note that the strategic behavior in our setting is not inherently malicious, but rather reflects rational decision-making based on cost, revenue, and expected trial outcomes. A good p-value threshold discourages agents with ineffective drugs from participating based on economic incentives: the costs of participation outweigh the potential benefits. This disincentives p-hacking insofar as data collection is costly. Our work also intersects with the literature on *contract theory*, which examines incentive alignment in the presence of private information [20, 21, 22, 23, 24]. Recent work by Min [25] applies contract theory to model FDA approval processes, analyzing how firms of different sizes choose between cheaper and more expensive trials. Further, Isakov et al. [26] employs Bayesian Decision Analysis (BDA) to optimize p-value thresholds by balancing Type I and Type II errors. Our empirical results leverage the statistical testing-based framework used by the FDA. Typically, this is established through either two successful controlled trials (with a p-value $< 0.05$) or a single robust multicenter trial (with a p-value $< 0.005$), as outlined in [7]. We simplify this in our analysis by only considering the former.

## 2 Model

**Preliminaries:** Consider a *principal* who must decide between approving or rejecting a product manufactured by an *agent* based on evidence provided by the agent. Let $X \in \{0, 1\}$ be a Bernoulli

random variable indicating whether the product was effective on a random instance. Let $\mu_0 = \mathbb{E}[X]$ denote the mean effectiveness of this product, with $\mu_0 \sim q$. While $\mu_0$ is private (known only by the agent), the distribution $q$ is assumed public (known to both the principal and agent). Let $\mu_b$ denote a baseline effectiveness (i.e. effectiveness of current products) and be treated as a constant: the principal's goal is to only approve products perceived to be at least as good as the baseline ($\mu_0 \geq \mu_b$), with ($\mu_0 - \mu_b$) denoted as *effect size*. An agent faces two possible decisions for their product. First, they can choose whether or not to participate in the approval process and engage with the principal. Not participating, equivalent to $n = 0$, incurs no cost and collects no revenue. If they participate, they must also decide on the number of samples $n \in [n_{min}, n_{max}]$ to collect and submit an *evidence set* $\mathcal{X}_n = (X_1, \ldots, X_n)$ to the principal. This incurs a fixed cost $c_0$ and a marginal per-sample cost $c$. Thus, $\text{cost}(n) = \mathbb{1}[n \neq 0](c_0 + cn)$. If their product is approved, they earn revenue $R$. Agents are assumed rational and act to maximize their expected utility – expected revenue minus cost (see Definition 2.2). We assume the revenue and cost parameters are known to the principal.

Connecting this to our running example, the FDA (the principal) decides on the approval of new drugs from pharmaceutical companies (the agents) based on whether the clinical trial data suggests them to be at least as effective as current alternatives on the market. If a firm follows through on the development of a new drug and participates in clinical trials, $c_0$ denotes any fixed costs herein, and $c$, the per-subject marginal cost of the trial. An FDA approval means a lifetime revenue $R$ for their new drug, while rejection means no revenue. As such, the firm only participates if its expected profit (utility) is non-negative, and chooses the samples $n$ to maximize this.

**Hypothesis Testing:** The agent's decision clearly hinges on *how* the principal uses the evidence set to evaluate effectiveness. We model this process on hypothesis testing given its ubiquity and relevance in settings like drug approval, manufacturing, public policy, and so on [7, 8, 9, 10]. Formally, let $\mathcal{H}_0 = \{\mu_0 \leq \mu_b\}$ be the *null hypothesis* (the product is less effective than baseline) and denote the alternative as $\mathcal{H}_1 = \{\mu_0 > \mu_b\}$. The principal uses a $p$-value threshold $\alpha$ to reject the null hypothesis and approve the product. To expand, given an evidence set $\mathcal{X}_n$, let $\hat{\mu}$ denote the empirical mean. Further, let $S_n$ denote the random sum of $n$ variables sampled from the baseline process with effectiveness $\mu_b$. Then the $p$-value is defined as the probability of observing outcomes at least as good as the evidence, conditioned on the null hypothesis: $p(\mu_0, n) = \Pr[S_n \geq n\hat{\mu}|\mathcal{H}_0]$ [2]. Observe that when the empirical mean of the evidence is worse than the baseline, the $p$-value will likely be higher than $\alpha$, leading the principal to reject; otherwise, when $p(\mu_0, n) \leq \alpha$, the principal will accept. For an evidence set with $n$ samples and a p-value $\alpha$, the *critical region* is the number of successes needed to reject the null-hypothesis and thus be approved. Since the sample outcomes are Bernoulli and the sum random variable follows a Binomial distribution, it is common to use a normal distribution $\phi$ to approximate the $p$-value and critical region:

$$z_{\alpha,n} \approx \left\{ k \in \mathbb{R} \,\middle|\, \int_k^\infty \phi(n\mu_b, n\mu_b(1-\mu_b)) \leq \alpha \right\} \approx n\mu_b + \Phi^{-1}(1-\alpha)\sqrt{n\mu_b(1-\mu_b)}$$

where $\Phi$ is the CDF of the standard normal and $\Phi^{-1}$, its inverse. Formalizing the principal's decision-making process means that we can now compile the probability that a product with effectiveness $\mu_0$ is approved (the randomness is over the evidence set), and thereby the agent's utility:

**Definition 2.1** (Pass Probability). *For a p-value threshold $\alpha$ and baseline effectiveness $\mu_b$, a product with effectiveness $\mu_0$ and evidence set $|\mathcal{X}_n| = n$ is approved with probability:*

$$Pass(\alpha, \mu_0, n) \triangleq \Pr[p(\mu_0, n) \leq \alpha] \approx 1 - \Phi\left( \frac{z_{\alpha,n} - n\mu_0}{\sqrt{n\mu_0(1-\mu_0)}} \right) \tag{1}$$

*Non-participation is considered equivalent to $n = 0$, and $Pass(\alpha, \mu_0, n = 0) \triangleq 0, \ \forall \mu_0, \alpha$.*

**Definition 2.2** (Agent Utility). *An agent with revenue $R$, cost parameters $(c_0, c)$, and a product with effectiveness $\mu_0 \sim q$ has the following utility for their participation/sample parameter $n$:*

$$u(\alpha, \mu_0, n) = R \cdot Pass(\alpha, \mu_0, n) - \mathbb{1}[n \neq 0](c_0 + cn) \tag{2}$$

While the agent aims to maximize their utility, the principal's goal is to choose a p-value threshold that minimizes some combination of their Type I and Type II errors, a standard desideratum in

---

[2]While $p$ value calculations depends on $\mu_b$, we drop this from function signatures since it is a constant.

hypothesis testing. In machine learning terminology, these are equivalent to minimizing the false positive (approving ineffective products) and false negative (not approving effective products) rates. These error components are defined with respect to the effectiveness distribution $q$, and we note that the principal may desire an arbitrary trade-off between the two error components. This principal loss and its objective are shortly defined in Definition 2.3.

**Game Theoretic Model:** It is evident that agents will choose $n$ strategically to maximize their utility, while the principal selects a p-value threshold $\alpha$ to minimize a *loss* function, which we will define in Definition 2.3. This naturally leads to a game-theoretic framework. In most regulatory settings (drug approval, manufacturing, etc), the principal must first communicate the acceptance criteria to all possible participants. Agents, on the other hand, can make their decision to participate and collect samples based on the revealed criteria. This interaction outlines a *Stackelberg Game*, an asymmetric model of strategic decision-making: the principal first commits to the p-value threshold, allowing agents to then make their optimal decision/best-response (participation decisions and number of samples) thereafter[3]. In such games, the core solution concept is the Stackelberg Equilibrium – the optimal principal strategy, *given* that downstream agents will best respond. We formally define the game and its details below:

**Definition 2.3** (Stackelberg Game in Strategic Hypothesis Testing)**.** *The principal-agent interactions in a hypothesis testing setting outline a Stackelberg game defined by the tuple $\mathcal{I} = (q, R, c, c_0, \mu_b)$. The best-response of an agent with effectiveness $\mu_0 \sim q$ when the principal commits to a p-value threshold $\alpha$ is: $n_{\mu_0}(\alpha) = \arg\max_n u(\alpha, \mu_0, n)$.*

*The principal in choosing a $p$ value threshold $\alpha$ suffers the following loss when agents best respond (we denote $Fail(\cdot) = 1 - Pass(\cdot)$ and $\lambda_{fp}, \lambda_{fn}$ are constants that scale the respective loss terms):*

$$\mathcal{L}(\alpha, I) = \lambda_{fp} \underbrace{\mathbb{E}_{\mu_0 \sim q} \left[ Pass(\alpha, \mu_0, n_{\mu_0}(\alpha)) | \mu_0 < \mu_b \right]}_{\substack{\text{False Positive (Type I error)} \triangleq FP(\alpha, q) \\ \text{Approval of ineffective products } (\mu_0 < \mu_b)}} + \lambda_{fn} \underbrace{\mathbb{E}_{\mu_0 \sim q} \left[ Fail(\alpha, \mu_0, n_{\mu_0}(\alpha)) | \mu_0 \geq \mu_b \right]}_{\substack{\text{False Negative (Type II error)} \triangleq FN(\alpha, q) \\ \text{Non-approval of effective products} (\mu_0 \geq \mu_b)}}$$

*The principal optimal strategy and the Stackelberg Equilibrium is to choose $\alpha^\star = \arg\min_\alpha \mathcal{L}(\alpha, \mathcal{I})$.*

In Section 3 and Section 4, we will investigate the agent's best response behavior as well as how the principal chooses p-value threshold affects the loss defined in Definition 2.3.

## 3 Agent Best Response

According to our strategic model, after the principal releases a $p$-value threshold $\alpha$, the utility-maximizing agent must decide (1) if they want to develop the product and participate in the approval process, and if so, (2) how many samples they ought to include in their submitted evidence set. Notice that this decision-making process is ex-ante and relies on the agent's belief in their product's effectiveness $\mu_0$. Intuitively, the agent intends to determine the optimal number of samples $n$ to maximize their ex-ante expected utility; if this is negative, they have no incentive to participate. Mathematically, participation is defined by $\mathbb{1}\left[ u(\alpha, \mu_0, n_{\mu_0}(\alpha)) \geq 0 \right]$, where $u(\alpha, \mu_0, n)$ is defined in Definition 2.2 and $n_{\mu_0}(\alpha)$ is the optimal number of samples. We now show that this can be efficiently computed by the agent. We sketch the proof below, with the full proof in Appendix B

**Theorem 3.1.** *For an instance $\mathcal{I}$ and a released p-value $\alpha$, an agent with effectiveness $\mu_0$ can compute their best-response (participation decision and number of samples) in $O(\log n_{max})$.*

*Proof Sketch.* By leveraging properties of the normal CDF that define the pass probability, we first show that when $\mu_0 < \mu_b$, the optimal $n = n_{min}$. For $\mu_0 \geq \mu_b$, we undertake a first and second-order analysis to partition the utility function into a constant number of intervals wherein it is convex or concave; the boundaries of these regions can be computed in constant time, and the optimal $n$ within each region requires at-most a binary search. Once the optimal $n$ is computed, it suffices to compute the passing probability and participate if the corresponding expected utility is non-negative. $\square$

We now prove that despite the agent being strategic and their participation and sampling decisions dynamically changing in both $\alpha$ and $\mu_0$, several natural and intuitive results still hold. These are

---

[3]Technically, our setting is a Bayesian Stackelberg Game since agents have hidden types (the effectiveness $\mu_0$), but the principal knows the distribution $q$.

instrumental to the analyses of the principal's optimal/equilibrium p-value threshold. We first show that the participation behaviour is monotonic in $\alpha$ (Lemma 3.1). The full proof is in Appendix B.

**Lemma 3.1.** *For an instance $\mathcal{I} = (q, R, c, c_0, \mu_b)$ and p-value threshold $\alpha$, if an agent with effectiveness $\mu_0$ participates in the statistical test, then any agent with belief $\mu_1 \geq \mu_0$ will also participate. Similarly, if an agent with belief $\mu_0'$ does not participate, neither will one with belief $\mu_1' < \mu_0'$.*

This monotonicity result immediately implies the existence of a *participation threshold* $\mu_\tau(\alpha)$ for any given p-value $\alpha$. That is, for any instance $\mathcal{I}$ and p-value $\alpha$, agents with effectiveness $\mu_0 \geq \mu_\tau(\alpha)$ will always participate, and those below will not. As we subsequently show, the participation threshold is a crucial concept in understanding the principal's decision – it determines the selection mechanism induced by the p-value threshold $\alpha$, shaping the set of participating agents based on their effectiveness. Consequently, understanding how $\mu_\tau(\alpha)$ changes as a function of the instance parameters $R, c, \alpha$ provides key insights into optimizing the principal's objective function. We now show that the participation threshold *decreases* as the p-value threshold $\alpha$ *increases*; further, it can be computed in log time and is agnostic to the effectiveness distribution $q$ (full proof in Appendix B).

**Definition 3.1** (Participation Threshold)**.** *For an instance $\mathcal{I}$ and a p-value $\alpha$, we denote $\mu_\tau(\alpha)$ as the participation threshold if it is optimal for agents with effectiveness $\mu_0 \geq \mu_\tau(\alpha)$ to participate, and for those with $\mu_0 < \mu_\tau(\alpha)$ to abstain.*

**Lemma 3.2.** *The participation threshold $\mu_\tau(\alpha)$ decreases in the p-value $\alpha$. Further, this threshold can be computed with $\varepsilon$ precision in $\mathcal{O}\left(\log\left(\frac{1}{\varepsilon}\right)\log(n_{max})\right)$.*

*Proof Sketch.* Since the passing probability of the hypothesis test increases with $\alpha$ for a fixed effect size, agents with lower beliefs are more likely to meet the passing condition as $\alpha$ increases. By the monotonicity of participation (Lemma 3.1), if an agent participates at $\mu_\tau(\alpha_1)$ under a p-value $\alpha_2 > \alpha_1$, then agents with beliefs greater than $\mu_\tau(\alpha_1)$ must also participate. This implies that the participation threshold under $\alpha_2$ must be at or below the threshold under $\alpha_1$, establishing that $\mu_\tau(\alpha)$ is non-increasing in $\alpha$. To determine $\mu_\tau(\alpha)$, we can discretize the effectiveness space into $\frac{1}{\varepsilon}$ intervals and run binary search. The algorithm and correctness rely on Lemma 3.1 and Theorem 3.1. $\qquad\square$

This formalizes the intuition that even under strategic sampling and participation decisions, as the principal increases the p-value threshold, namely when it's "easier" to pass the statistical test, agents with lower effectiveness (who can still be better than baseline) will find it beneficial to participate in the approval process. The monotonicity of the agent participation decision also means that the agent's utility is increasing in $\mu_0$. This is formalized below, with the proof in Appendix B.

**Lemma 3.3.** *For any given p-value $\alpha$, the agent's utility under optimal number of samples $u(\alpha, \mu_0, n_{\mu_0}(\alpha))$ increases in their effectiveness $\mu_0$.*

## 4   Principal Equilibrium and Loss Dynamics

The principal's strategic choice is to set the $p$-value threshold $\alpha$. Given the agent's best response behavior, how should the principal select $\alpha$ to minimize their loss, as defined in Definition 2.3? The loss captures the trade-off between two risks: approving ineffective products (false positives) and failing to approve effective ones (false negatives). A product can fail to be approved either by explicit rejection after participating in the approval process, or by the agents opting not to participate. Since the $p$-value threshold $\alpha$ influences the participation decision (Lemma 3.2), this becomes an implicit factor in the loss function. The principal's challenge is to set $\alpha$ optimally: a low $\alpha$ reduces false positives but may discourage worthy candidates from participation, increasing false negatives; conversely, a high $\alpha$ boosts participation but raises the risk of approving ineffective products. Optimizing $\alpha$ requires a careful balance between these risks while accounting for agents' strategic behaviour.

We begin by re-expressing the cumulative loss conditioned on the participation decision induced by $\alpha$ for a given agent, a decision entirely captured by the condition $\mu_0 \geq \mu_\tau(\alpha)$:

$$\mathcal{L}(\alpha, \mathcal{I}) = \lambda_{fp} \mathop{\mathbb{E}}_{\mu_0 \sim q} [\text{Pass}(\alpha, \mu_0, n_{\mu_0}(\alpha))|\mu_0 < \mu_b, \mu_0 \geq \mu_\tau(\alpha)]P(\mu_0 \geq \mu_\tau(\alpha)|\mu_0 \leq \mu_b) \quad (\text{FP}_{\text{particip}})$$

$$+ \lambda_{fp} \mathop{\mathbb{E}}_{\mu_0 \sim q} [\text{Pass}(\alpha, \mu_0, n_{\mu_0}(\alpha))|\mu_0 < \mu_b, \mu_0 < \mu_\tau(\alpha)]P(\mu_0 < \mu_\tau(\alpha)|\mu_0 \leq \mu_b) \quad (\text{FP}_{\text{abstain}})$$

$$+ \lambda_{fn} \mathop{\mathbb{E}}_{\mu_0 \sim q} [\text{Fail}(\alpha, \mu_0, n_{\mu_0}(\alpha))|\mu_0 \geq \mu_b, \mu_0 \geq \mu_\tau(\alpha)]P(\mu_0 \geq \mu_\tau(\alpha)|\mu_0 \geq \mu_b) \quad (\text{FN}_{\text{particip}})$$

$$+ \lambda_{fn} \mathop{\mathbb{E}}_{\mu_0 \sim q} [\text{Fail}(\alpha, \mu_0, n_{\mu_0}(\alpha)) | \mu_0 \geq \mu_b, \mu_0 < \mu_\tau(\alpha)] P(\mu_0 < \mu_\tau(\alpha) | \mu_0 \geq \mu_b) \quad (\text{FN}_{\text{abstain}})$$

$$= \text{FP}_{\text{particip}}(\alpha, \mathcal{I}) + \text{FP}_{\text{abstain}}(\alpha, \mathcal{I}) + \text{FN}_{\text{particip}}(\alpha, \mathcal{I}) + \text{FN}_{\text{abstain}}(\alpha, \mathcal{I})$$

Note that the overall false positive $\text{FP}(\alpha, \mathcal{I}) = \text{FP}_{\text{particip}}(\alpha, \mathcal{I}) + \text{FP}_{\text{abstain}}(\alpha, \mathcal{I})$ and similarly, the overall false negative $\text{FN}(\alpha, \mathcal{I}) = \text{FN}_{\text{particip}}(\alpha, \mathcal{I}) + \text{FN}_{\text{abstain}}(\alpha, \mathcal{I})$. Further, observe that when the agent does not participate, their probability of passing the test is 0 – therefore, $\text{FP}_{\text{abstain}} = 0$ at all times. Thus, the cumulative loss is fully specified by $\mathcal{L}(\alpha, \mathcal{I}) = \text{FP}_{\text{particip}} + \text{FN}_{\text{particip}} + \text{FN}_{\text{abstain}}$. We will now analyze the properties of these components as a function of $\alpha$. Remarkably, despite the complex interplay between the agent's strategic participation and the principal's decision, the loss terms exhibit consistent monotonic behavior within regions segmented by a *critical threshold* $\widehat{\alpha}$ – intuitively, this is the p-value wherein the participation threshold $\mu_\tau(\widehat{\alpha}) = \mu_b$ (see Definition 4.1).

**Definition 4.1** (Critical p-value $\widehat{\alpha}$). *The critical p-value threshold $\widehat{\alpha}$ is the p-value at which the participation threshold (Definition 3.1) is equal to the baseline effectiveness – $\mu_\tau(\widehat{\alpha}) = \mu_b$. This quantity is agnostic to the effectiveness distribution q and scaling parameters $(\lambda_{fp}, \lambda_{fn})$.*

Before our detailed analysis, we highlight that the forthcoming results depend on a core observation – for effective agents $(\mu_0 \geq \mu_b)$, the probability of passing the statistical test increases as the p-value $\alpha$ increases. While this is intuitive when the participation and the number of samples used $(n)$ are fixed, when agents are strategic and dynamically change these decisions, it is trickier. For example, if an increased $\alpha$ led the agent to use fewer samples due to high marginal cost relative to the increase in passing probability and thus revenue (this is indeed common as the p-value becomes high), the pass probability could decrease. The lemma below highlights that despite the complicated dynamics of optimal samples, $n_{\mu_0}(\alpha)$, the passing probability is always non-decreasing in $\alpha$. The proof is technical and appears in Appendix C, but we sketch the key ideas below.

**Lemma 4.1.** *For an instance $\mathcal{I}$ and an agent with effectiveness $\mu_0 \geq \mu_b$, the probability of passing – $\text{Pass}(\alpha, \mu_0, n_{\mu_0}(\alpha))$ – is non-decreasing in $\alpha$ when the agent participates and uses $n_{\mu_0}(\alpha)$ samples.*

*Proof Sketch.* We prove that for $\mu_0 \geq \mu_b$, we have $\frac{\partial}{\partial \alpha} \text{Pass}(\alpha, \mu_0, n_{\mu_0}(\alpha)) \geq 0$. The total effect is decomposed into a direct effect, capturing the change in pass probability purely due to $\alpha$, and an indirect effect, which accounts for changes in the agent's optimal sample size $n_{\mu_0}(\alpha)$. For the direct effect, an increase in $\alpha$ reduces the critical threshold $\Phi^{-1}(1 - \alpha)$, making it easier for agents to pass the test. For the indirect effect, the agent adjusts their optimal sample size $n_{\mu_0}(\alpha)$ based on the value of $\alpha$. We compute $\frac{\partial \text{Pass}}{\partial n}$ and apply the Implicit Function Theorem to determine $\frac{dn(\alpha)}{d\alpha}$, which incorporates the agent's optimal behavior. The second derivative of the pass probability with respect to $n$ is shown to be negative, ensuring the utility is concave with respect to sample size. This concavity guarantees a unique optimal sample size $n_{\mu_0}(\alpha)$, which varies predictably with $\alpha$. Finally, we show that the combination of the direct and indirect effects results in the total derivative being non-negative, confirming that the pass probability is monotonic with respect to $\alpha$. $\square$

## 4.1 False Positive Loss

Since $\text{FP}_{\text{abstain}} = 0$ (non-participation means the probability of passing is 0), the overall false positive is purely determined by $\text{FP}_{\text{particip}}$. We now show below that for $\alpha \leq \widehat{\alpha}$, $\text{FP}_{\text{particip}} = 0$, while for $\alpha > \widehat{\alpha}$, this is increasing. We set the scaling factor $\lambda_{fp} = 1$ since it does not affect the analysis.

**Theorem 4.1.** *The overall false positive loss, $FP(\alpha, \mathcal{I}) = FP_{particip}(\alpha, \mathcal{I})$. Further, this quantity is 0 for $\alpha < \widehat{\alpha}$, and non-decreasing in $\alpha$ for $\alpha \geq \widehat{\alpha}$.*

*Proof.* It suffices to prove the property for $\text{FP}_{\text{particip}}$. First, consider an $\alpha < \widehat{\alpha}$. We know from Lemma 3.1 that the participation threshold is decreasing in $\alpha$. Thus, since $\mu_\tau(\widehat{\alpha}) = \mu_b$, it means that our given $\alpha$, $\mu_\tau(\alpha) > \mu_b$. In other words, any agents participating under this p-value are effective (above baseline) and do not contribute to the false positive.

Next, consider $\alpha \geq \widehat{\alpha}$. We know from above that the participation threshold now is below $\mu_b$. Observe that for any $\alpha$, we can write the false positive rate as follows:

$$\text{FP}_{\text{particip}} = \frac{P[\mu_\tau(\alpha) \leq \mu_0 \leq \mu_b]}{P[\mu_0 \leq \mu_b]} \int_{\mu_\tau(\alpha)}^{\mu_b} \text{Pass}(\alpha, \mu_0, n_{\mu_0}(\alpha)) P[\mu_0 | \mu_\tau(\alpha) \leq \mu_0 \leq \mu_b] d\mu_0$$

$$= \frac{1}{P[\mu_0 \leq \mu_b]} \int_{\mu_\tau(\alpha)}^{\mu_b} \text{Pass}(\alpha, \mu_0, n_{\mu_0}(\alpha)) q(\mu_0) d\mu_0$$

Note that $\frac{1}{P[\mu_0 \leq \mu_b]}$ is a constant and as $\alpha$ increases to $\alpha'$, $\mu_\tau(\alpha') < \mu_\tau(\alpha)$, meaning we integrate over a larger region. The integrand itself is always positive, and for every $\mu_0 \geq \max(\mu_\tau(\alpha), \mu_\tau(\alpha')) = \mu_\tau(\alpha)$, the pass probability has increased (Lemma 4.1). Thus, $\text{FP}_{\text{particip}}$ is non-decreasing in $\alpha$. □

### 4.2 False Negative Loss

We next consider the components of the false negative loss: $\text{FN}_{\text{particip}}$ and $\text{FN}_{\text{abstain}}$. We first establish the monotonicity properties of $\text{FN}_{\text{abstain}}$, which is exactly equal to the proportion of effective agents not participating since not participating means that $\text{Pass}(\cdot) = 0$, implying $\text{Fail}(\cdot) = 1$. In other words, $\text{FN}_{\text{abstain}} = P(\mu_0 < \mu_\tau(\alpha)|\mu_0 \geq \mu_b)$. Once more, segmenting the p-value space by $\widehat{\alpha}$ is crucial to establishing this relationship. As before, we set $\lambda_{fn} = 1$ since it does not affect the analysis.

**Proposition 4.1.** *The non-participation false negative loss, $\text{FN}_{\text{abstain}}(\alpha, I)$ is non-increasing in $\alpha$ for $\alpha \leq \widehat{\alpha}$, and is 0 for any $\alpha \geq \widehat{\alpha}$.*

*Proof.* Consider $\alpha_1, \alpha_2$ where $\alpha_2 \geq \alpha_1$, where both $\alpha_1, \alpha_2 \leq \widehat{\alpha}$. Thus, we first need to show that: $P(\mu_0 < \mu_\tau(\alpha_1)|\mu_0 \geq \mu_b) \geq P(\mu_0 < \mu_\tau(\alpha_2)|\mu_0 \geq \mu_b)$. We know from the monotonicity of the threshold belief (Lemma 3.1), $\mu_\tau(\alpha_2) \leq \mu_\tau(\alpha_1)$. Let $Q_{\geq \mu_b}$ denote the CDF of the effectiveness distribution, conditioned on the event $\mu_0 \geq \mu_b$. Then $Q_{\geq \mu_b}(\mu_\tau(\alpha)) = P[\mu_0 \leq \mu_\tau(\alpha)|\mu_0 \geq \mu_b]$. It is then immediate that $Q_{\geq \mu_b}(\mu_\tau(\alpha_2)) \leq Q_{\geq \mu_b}(\mu_\tau(\alpha_1))$. Lastly, for $\alpha \geq \widehat{\alpha}$, we note that by definition of $\widehat{\alpha}$ and Lemma 3.1: $\mu_\tau(\alpha) \leq \mu_b$. In other words, it is optimal for all effective agents to participate, meaning $\text{FN}_{\text{abstain}} = 0$ in this regime. □

One might expect to show a similar result for $\text{FN}_{\text{particip}}$ and thereby conclude the cumulative behavior of the false negative loss. Indeed, when $\alpha \geq \widehat{\alpha}$, the participation threshold is smaller than the baseline $\mu_\tau(\alpha) < \mu_\tau(\widehat{\alpha}) = \mu_b$, which means all agents whose prior is greater than $\mu_b$ will participate, thus $\Pr[\mu_0 \geq \max(\mu_\tau(\alpha), \mu_b)|\mu_0 \geq \mu_b] = 1$. The FN under participation loss simplifies to

$$\text{for } \alpha \geq \widehat{\alpha}: \quad \text{FN}_{\text{particip}}(\alpha, I) = \mathop{\mathbb{E}}_{\mu_0 \sim Q}[1 - \text{Pass}(\alpha, \mu_0, n_{\mu_0}(\alpha))|\mu_0 \geq \mu_b] \tag{3}$$

Following directly from the monotonicity of the passing probability (Lemma 4.1), this implies that $\text{FN}_{\text{particip}}$ decreases as a function of $\alpha$ when $\alpha \geq \widehat{\alpha}$. We formally state this below:

**Proposition 4.2.** *$\text{FN}_{\text{particip}}(\alpha, I)$ is non-increasing in $\alpha$ for all $\alpha \geq \widehat{\alpha}$.*

When $\alpha < \widehat{\alpha}$, however, $\text{FN}_{\text{particip}}$ displays a much more nuanced behaviour – indeed, in Section 5, we show that this component is not monotonic and can be increasing, decreasing, or both in the $\alpha \leq \widehat{\alpha}$ region. This may suggest that, unlike the false positive setting, little can be said about the combined false negative effect ($\text{FN}_{\text{particip}} + \text{FN}_{\text{abstain}}$). As we show in Theorem 4.2, however, this is not true, and we directly establish the monotonicity of the combined false negative rate when the p-value domain is segmented by $\widehat{\alpha}$. The result is technical and the full proof is given in Appendix C.

**Theorem 4.2.** *The overall false negative loss, $\text{FN}(\alpha, \mathcal{I}) = \text{FN}_{\text{particip}}(\alpha, \mathcal{I}) + \text{FN}_{\text{abstain}}(\alpha, \mathcal{I})$, is non-increasing in $\alpha$ on both sides of $\widehat{\alpha}$, that is, it is not increasing for $\alpha \leq \widehat{\alpha}$, and for all $\alpha \geq \widehat{\alpha}$.*

### 4.3 Characterizing $\alpha^*$ – The Optimal p-value Threshold

| | $\text{FP}_{\text{particip}}$ | $\text{FP}_{\text{abstain}}$ | **Overall FP** | $\text{FN}_{\text{particip}}$ | $\text{FN}_{\text{abstain}}$ | **Overall FN** |
|---|---|---|---|---|---|---|
| $\alpha \leq \widehat{\alpha}$ | 0 | 0 | **0** | – | Non-Inc | **Non-Inc** |
| $\alpha > \widehat{\alpha}$ | Non-Dec | 0 | **Non-Dec** | Non-Inc | 0 | **Non-Inc** |

Table 1: Summary of the loss dynamics as $\alpha$ increases. $\widehat{\alpha}$ is defined in Definition 4.1. We comment on $\text{FN}_{\text{particip}}$ in the $\alpha \leq \widehat{\alpha}$ regime in Section 5. See figure 1 for this dynamic visualized on an instance.

We summarize our analysis of the different loss components in the table above. These results lead to a succinct characterization of how the principal should use their equilibrium $\alpha^*$. Observe that for any $\alpha \leq \widehat{\alpha}$, the overall false positive is 0, while the overall false negative rate is decreasing in $\alpha$. This

immediately implies that $\widehat{\alpha}$ is weakly better than any $\alpha \leq \widehat{\alpha}$, and that $\alpha^*$ must be at least as large as $\widehat{\alpha}$! This observation and the value of $\widehat{\alpha}$ are agnostic to both the effectiveness distribution $q$ and the loss scaling constants $\lambda_{fp}, \lambda_{fn}$. This is a powerful characterization. While the true optimal $\alpha^*$ may be larger, the principal suffers additional false positives here while lowering the false negatives. In settings like drug approvals and manufacturing, where the cost of false-positives is very high – $\lambda_{fp} \gg \lambda_{fn}$ – the optimal may indeed be close to $\widehat{\alpha}$.

In recommending values close to $\widehat{\alpha}$ as optimal, the question of computation arises. Fortunately, since by definition at $\widehat{\alpha}$, $\mu_\tau(\widehat{\alpha}) = \mu_b$, and the participation threshold $\mu_\tau(\alpha)$ can be computed to $\varepsilon$ accuracy in $\mathcal{O}(\log \frac{1}{\varepsilon} \log(n_{max}))$ time, an $\widehat{\alpha}$ can be efficiently computed to $\varepsilon$ accuracy in $\mathcal{O}(\log^2 \frac{1}{\varepsilon} \log(n_{max}))$. This follows from discretizing the p-value domain into $\frac{1}{\varepsilon}$ intervals and running binary search, leveraging the monotonicity of the participation threshold Lemma 3.2.

**Theorem 4.3.** *For any instance $\mathcal{I}$, the principal's optimal p-value $\alpha^*$ satisfies $\alpha^* \geq \widehat{\alpha}$, where $\widehat{\alpha} | \mu_\tau(\widehat{\alpha}) = \mu_b$. Further, an $\varepsilon$ approximation to an $\widehat{\alpha}$ such that $\mu_\tau(\widehat{\alpha}) \in [\mu_b \pm \varepsilon]$ can by computed in $\mathcal{O}(\log^2 \frac{1}{\varepsilon} \log(n_{max}))$.*

## 5  Experimental Studies

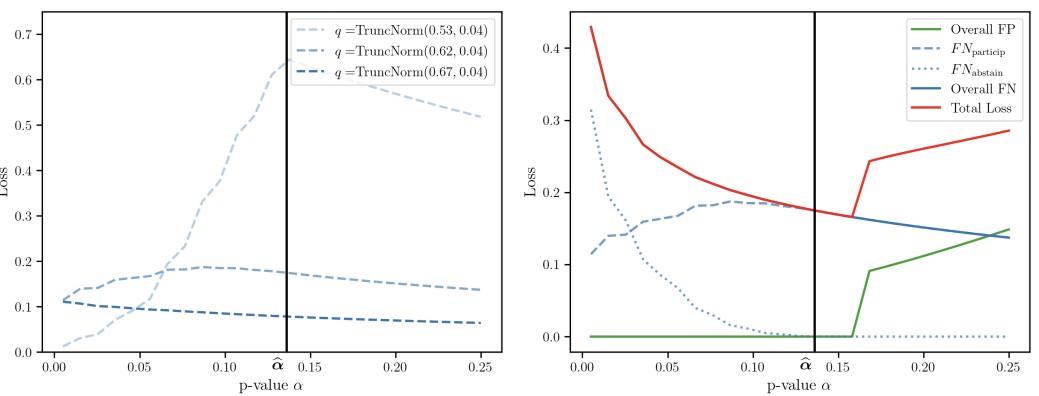

Figure 1: On the left, we plot the $FN_{particip}$ losses for different effectiveness distributions $q$. On the right, we plot all the different loss components for one of these distributions with $\lambda_{fp} = \lambda_{fn} = 1$

### 5.1  False Negative Rate Among Participating Agents

Our technical analysis in the preceding section accounts for all components of the loss except one: $FN_{particip}$ for $\alpha \leq \widehat{\alpha}$. For a principal unaware of the strategic implications of their chosen significance level ($\alpha$)—particularly that it may discourage participation by agents with above-baseline effectiveness—this may be the only false negative they observe. At first glance, it seems increasing $\alpha$ should reduce this: the test becomes easier to pass, so effective agents should be more likely to succeed. Theorem 4.2 also aligns with this intuition since the overall FN rate decreases with $\alpha$.

However, $FN_{particip}$ behaves more intricately. We highlight this with an example. Consider $\mu_b = 0.5$ and a truncated normal distribution $q$ between 0.4 and 0.7 with standard deviation 0.04. First, suppose the mean of $q$ is 0.53, meaning a large mass of effective agents are only slightly better than baseline. As $\alpha$ increases from $\alpha_1$ to $\alpha_2$, the participation threshold falls (by Lemma 3.1), causing a large mass of these slightly-effective agents to switch from abstaining to participating. However, due to their small effect size and samples being costly, these new participants pass with a lower probability than those who also participated under $\alpha_1$. Their inclusion in the FN metric at $\alpha_2$ outweighs increased pass probability among previously participating agents, increasing the FN rate under $\alpha_2$.

This effect is not monotonic and depends on the distribution. If $q$'s mean rises to 0.62, the same initial increase in FN rate occurs in the small $\alpha$ regime. For higher values of $\alpha$, however, most effective agents are already participating, and few agents remain that are both slightly better than baseline and could switch; as $\alpha$ increases, the effect is dominated by the increasing pass probability of those already participating, decreasing the $FN_{particip}$. As shown in Figure 1, $FN_{particip}$ increases and

then decreases. In selecting a higher mean (0.67), $\text{FN}_{\text{particip}}$ is always decreasing. Overall, this rich behavior is fundamentally due to participation and evidence set size being a strategic decision.

## 5.2 Analysis of $\widehat{\alpha}$ in Drug Approvals

We now conduct a case study of our strategic hypothesis testing model by considering our running example: drug approvals. Despite much of the data in this setting being proprietary, we use public sources to capture relevant metrics for three classes of drugs: oncology, vaccines, and cardiovascular. ProRelix [27] mentions the per-participant expense ($c$) of vaccine testing to be $\sim \$50,000$, while oncology and cardiovascular trials being around $\sim \$128,000$ and $\sim \$136,000$ respectively. The fixed expenditure ($c_0$) for clinical trials, regulatory approvals, and R&D, as well as the lifetime revenue ($R$) vary widely, even within drug categories. Prasad and Mailankody [28] highlights the median fixed expenditure to be around \$650 million for oncology drugs, although in the extremes it can be as low as \$150 million or higher than a billion. The median four year revenue to be around 1.6 Billion; data on lifetime revenue is limited, but [29, 30] suggest it can be around 10-15 billion, with blockbuster drugs above 50 billion [14]. Analysis from Bhatt et al. [16] gives ranges of \$74 - \$183 million for fixed costs of Cardiovascular drugs with a median of \$141 million. Rashid and Chandel [31] suggests the corresponding revenue of approved drugs here to be between under a billion to over 10 billion, with a median of around \$3.5 billion. For vaccines, Sertkaya et al. [32] outlines revenues between 6.9 billion to 36.9 billion for blockbusters, with median fixed costs around \$886 million. In the extremes, it can be below \$100 million or above a billion. We present this rough data in Section 5.2.

| Drug Category | Revenue if approved (R) | Fixed Cost ($c_0$) | Cost Per Sample ($c$) |
|---|---|---|---|
| Oncology | $[1,500 - 50,000]$ | $(648); [150 - 1,000]$ | 0.136 |
| Cardiovascular | $(3,560); [1,000 - 10,000]$ | $(141); [74 - 183]$ | 0.128 |
| Vaccine | $[6,900 - 36,900]$ | $(886); [100 - 1,000]$ | 0.05 |

Table 2: Rough costs and revenue by drug category. The median value, where available, is presented in parentheses. All numbers in millions USD, for US-based development.

Figure 2: $\widehat{\alpha}$ vs Revenue - Oncology      Figure 3: $\widehat{\alpha}$ vs Revenue - Cardiovascular

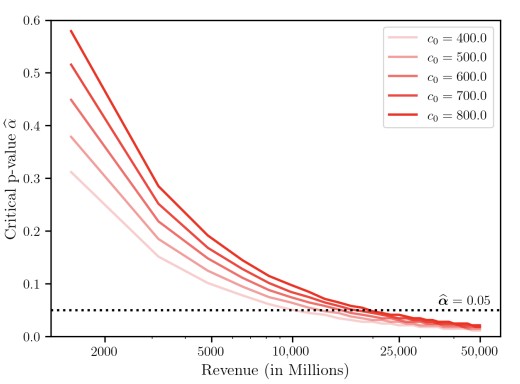 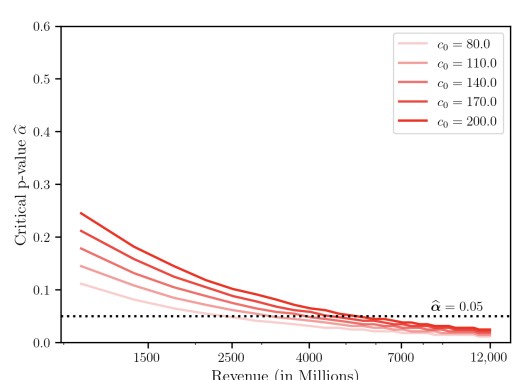

Our goal is to understand what $\widehat{\alpha}$, the critical p-value that weakly dominates any lower p-value , looks like in the drug approval setting. Due to the variability of fixed costs $c_0$ and revenue $R$ as compared to per-sample-cost $c$, we fix $c$ and plot $\widehat{\alpha}$ for different revenue and fixed costs. These ranges are centered around the median values (where available) gathered in the table. We plot the figures for oncology and cardiovascular categories above (see Figure 2 and 3); the vaccines figure is in Appendix E)[4]. Each plot also displays the $\alpha = 0.05$ boundary, a commonly used p-value threshold by the FDA [34].

The results suggest that our strategic model is roughly capturing the economic dynamics of drug approvals. The in-practice used p-value of $0.05$ is the ideal choice at around the median revenue for

---

[4]The experiments were conducted using a MacBook with only CPU resources and the NumPy package [33].

each drug category. Within our leader-follower dynamic, the FDA is committing to this threshold, first suggests that pharmaceuticals are choosing their parameters (samples collected, price, etc) to ensure profitability. Conversely, if regulatory bodies wish for decreased drug prices with the fixed costs being immutable, they ought to specify a higher p-value threshold, as the current value may dissuade effective but low-revenue drugs. In oncology, for instance, the critical threshold $\widehat{\alpha}$ for around $5 billion revenue is between 0.1 and 0.2. Given our analysis, such a value does not induce any more false positives than more stringent ones (Theorem 4.1), but lowers the false negative rate. However, one may argue that choosing a higher p-value would imply that those with much higher revenue (say 25 billion) will be approved even when they are ineffective since their $\widehat{\alpha}$ is lower. Practically, this is unlikely to happen since it is rare that an ineffective drug, albeit FDA-approved, would achieve such blockbuster revenue, as the market and post-market surveillance would generally expose inefficacy. Conversely, choosing p-values stricter than $\widehat{\alpha}$ hamstrings low-revenue but effective drugs; this is a real concern for orphan drugs or low-cost therapeutics where margins are much slimmer.

## 6   Discussion

This work spiritually extends the growing literature on strategic machine learning to hypothesis testing, a ubiquitous method used in regulatory approvals. Given the costs of data collection, firms may strategically choose smaller trials or opt out entirely if the expected revenue, shaped by approval probabilities under the chosen threshold, does not justify the expense. By modeling this dynamic within a principal-agent framework, our work uncovers a systematic connection between the regulator-specified $p$-value and the resulting firm decision. This allows us to give an interpretable characterization of the regulator's optimal $p$-value, which minimizes false positive and negative error rates under such strategic consideration. We validate the findings of our model with real cost, revenue, and approval data from the US pharmaceutical sector and derive two key policy insights. First, the commonly used FDA $p$-value of 0.05 is fairly strict given development costs, requiring firms to extract large revenue from approved drugs for economic viability. This matches empirical evidence from the US pharmaceutical market. Second, to decrease drug costs while maintaining minimal error rates, regulators can either selectively increase the $p$-value threshold or give subsidies for running trials; our model gives guidance on how to adjust either lever.

Our main results rely on all parties knowing the cost and revenue parameters $(c_0, c, R)$ – this determines agent participation and the principal's optimal threshold $\hat{\alpha}$[5]. While these quantities may in practice be estimated, our qualitative conclusions are robust to small estimation errors. In Appendix D, we discuss how small perturbations in the cost and revenue parameters affect the agents' participation thresholds and the principal's optimal choice of $\hat{\alpha}$, and argue that these perturbations have only a limited impact and leave the qualitative behavior of the model unchanged.

## Acknowledgments and Disclosure of Funding

Yiling Chen was partially supported by the National Science Foundation under grant IIS 2147187.

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

# A   Notation Table

| Symbol | Usage |
|--------|-------|
| $X \subset \{0,1\}$ | Bernoulli random variable indicating whether a product was effective on a random instance |
| $q$ | PDF of the effectiveness distribution |
| $Q$ | CDF of the effectiveness distribution $Q$. |
| $\mu_0$ | product effectiveness |
| $\mu_b$ | baseline effectiveness |
| $\sigma_0$ | variance of a product's effectiveness: $\sigma_0 = \sqrt{(1-\mu_0)\mu_0}$ |
| $\sigma_b$ | variance of a baseline product's effectiveness: $\sigma_b = \sqrt{(1-\mu_b)\mu_b}$ |
| $c_0$ | fixed cost of running a clinical trial |
| $c$ | per-sample cost |
| $R$ | revenue upon approval |
| $\alpha$ | p-value threshold used by the principal |
| $p(\mu_0, n; \mu_b)$ | the probability of observing outcomes at least as good as the evidence conditioned on the null hypothesis $\mu_b$ |
| $z_{\alpha,n}$ | The critical region given $n$ samples and p-value $\alpha$ |
| $\mathrm{Pass}(\alpha, \mu_0, n)$ | the probability that a product with effectiveness $\mu_0$ is approved |
| $u(\alpha, \mu_0, n)$ | utility for an agent with effective size $\mu_0$ and number of samples $n$ |
| $n_{\mu_0}(\alpha)$ | optimal number of samples given p-value threshold $\alpha$ and effectiveness $\mu_0$ |
| $\mu_\tau(\alpha)$ | The participation threshold belief under a p-value threshold $\alpha$ |
| $\alpha^*$ | The critical p-value threshold such that $\mu_\tau(\alpha^*) = \mu_b$ |

Table 3: Primary Notation

# B  Omitted proofs in Section 3

**Proof of Theorem 3.1**

*Proof.* We first define the following variables for brevity. Let $\Delta_\mu = \mu_0 - \mu_b$ denote the effect size, $\sigma_0 = \sqrt{\mu_0(1 - \mu_0)}$ denote the standard deviation of the true process and $\sigma_b = \sqrt{\mu_b(1 - \mu_b)}$ the standard deviation of the baseline Bernoulli variable. We use $\phi(z)$ to denote the standard normal, and $\Phi(z)$ to denote its CDF. Lastly, let $d_\alpha = \Phi^{-1}(1 - \alpha)$ denote the $1 - \alpha$ percentile of the standard normal. Recall that we work under the normal approximation of the binomial distribution where the critical threshold is:

$$z'_{\alpha,n} = \left\{ k \in \mathbb{R} \,\middle|\, \int_k^\infty \mathcal{N}(n\mu_b, n\mu_b(1 - \mu_b)) \right\} = n\mu_b + \Phi^{-1}(1-\alpha)\sqrt{n\mu_b(1 - \mu_b)} = \mu_b n + d_\alpha \sigma_b \sqrt{n} \tag{4}$$

Thus, we may write the pass probability as follows, from which the expected utility expression follows:

$$\text{Pass}(\cdot) = 1 - \Phi\left( \frac{z'_{n,\alpha} - n\mu_0}{\sqrt{n\mu_0(1 - \mu_0)}} \right) = 1 - \Phi\left( \frac{n\mu_b + d_\alpha \sqrt{n\mu_b(1 - \mu_b)} - n\mu_0}{\sqrt{n\mu_0(1 - \mu_0)}} \right)$$

$$= 1 - \Phi\left( \frac{d_\alpha \sigma_b}{\sigma_0} - \frac{\Delta_\mu \sqrt{n}}{\sigma_0} \right)$$

$$\implies u(n; \alpha, \mu_0, \mu_b) = R - R\Phi\left( \frac{d_\alpha \sigma_b}{\sigma_0} - \frac{\Delta_\mu \sqrt{n}}{\sigma_0} \right) - cn - c_0$$

Observe that when the drug is not effective, i.e. $\mu_0 < \mu_b$, then the argument to $\Phi()$ increases in $n$. Hence, the passing probability, and thus the expected revenue, decreases with increasing $n$. It is evident that the cost also increases in $n$. Thus, when the underlying drug is not effective, it is optimal for the agent to choose the smallest value $n_{min}$ possible. Computing the pass probability oracle with $n_{min}$, they can determine their maximum expected utility and choose to participate if this is positive. In other words, when $\mu_0 < \mu_b$, a constant amount of computation suffices to determine the optimal participation decision.

We next turn to the more interesting case where $\mu_0 \geq \mu_b$. Our goal is to efficiently search the sample space by leveraging structural properties of the utility function. Letting $v = \frac{d_\alpha \sigma_b}{\sigma_0} - \frac{\Delta_\mu \sqrt{n}}{\sigma_0}$, the derivative of the utility function with respect to $n$ is as follows:

$$\frac{\partial u}{\partial n} = -R\frac{\partial \Phi}{\partial v}\frac{\partial v}{\partial n} - c \tag{5}$$

$$= \phi(v)\frac{R\Delta_\mu}{2\sigma_0\sqrt{n}} - c = \frac{R\Delta_\mu}{2\sigma_0\sqrt{n}}\phi\left( \frac{d_\alpha \sigma_b}{\sigma_0} - \frac{\Delta_\mu \sqrt{n}}{\sigma_0} \right) - c \tag{6}$$

The first term is always positive, and since the image of $\phi()$ is always above 0. As $n$ increases, this first term tends to 0, and the derivative is dominated by the second term and becomes negative. In other words, increasing $n$ only increases utility to a point. To precisely understand this characteristic, consider the second derivative of the utility function:

$$\frac{\partial^2 u}{\partial n^2} = \frac{-R\Delta_\mu}{2\sigma_0 n^{3/2}}\phi(v) + \frac{R\Delta_\mu^2}{2\sigma_0^2 n}v\phi(v) = \frac{R\phi(v)\Delta_\mu}{2\sigma_0 n^{3/2}}\left[ \frac{\Delta_\mu v}{2\sigma_0}\sqrt{n} - 1 \right] \tag{7}$$

The sign of the second derivative, and thus the concavity/convexity properties of the utility function depend solely on $\left[ \frac{\Delta_\mu v}{2\sigma_0}\sqrt{n} - 1 \right]$ since the term multiplying it is always positive. We now expand it by plugging in the definition of $v$:

$$\left[ \frac{\Delta_\mu v}{2\sigma_0}\sqrt{n} - 1 \right] = \frac{-\Delta_\mu}{2\sigma_0^2}n + \frac{d_\alpha \Delta_\mu \sigma_b}{2\sigma_0^2}\sqrt{n} - 1 = \frac{-\Delta_\mu}{2\sigma_0^2}t^2 + \frac{d_\alpha \Delta_\mu \sigma_b}{2\sigma_0^2}t - 1 \tag{8}$$

where we substitute in $t = \sqrt{n}$. Observe that this is a negative quadratic expression with only positive real root being meaningful (since $\sqrt{n}$ cannot be negative). The roots $n_1, n_2$ can easily be computed by applying the quadratic formula to the instance parameters. We categorize the outcome as follows:

1. No positive roots $\implies$ the function is always *concave*
2. One positive root at $n_1 \implies$ the function is *convex* over $[0, n_1]$ and *concave* over $[n_1, \infty)$.
3. Two positive root at $n_1, n_2$ with $n_2 > n_1 \implies$ that over $[0, n_1]$ the function is *concave*, over $[n_1, n_2]$ is is *convex* and over $[n_2, \infty)$ it is *concave*.

We can consider the problem of optimizing $n$ over each of these convex/concave regions defined by the roots. Note that a convex function always attains its maximum on the boundary. Thus, for a the convex interval, it suffices to just check the boundary points, of which there are only two. At the optimal $n$ for a concave interval, the first derivative is always 0, or it is a boundary point. Since the latter case is similar to the first one, it suffices to find where the first derivative is 0, if it exists. Since the first derivative is always increasing (monotonic) and we need to find the $x$-intercept, a binary search suffices: starting in the middle of the interval, increase $n$ if the first derivative is positive, and decrease it if negative. This makes at most $\log n_{max}$ calls. It is immediate that taking max over the best $n$ from each of the at most three yields the globally optimal $n_{\mu_0}(\alpha)$. The agent can compute the corresponding passing probability and utility and decide to participate if this is positive. $\qquad\square$

**Proof of Lemma 3.1**

*Proof.* We start with the first direction. If an agent with prior belief $\mu_0$ participates with $n$ samples, then by individual rationality, they must have non-negative utility. That is, $\Pr[p(n, \mu_0; \mu_b) \leq \alpha] \cdot R \geq (c \cdot n + c_0)$, where $(c \cdot n + c_0)$ is the cost. Now consider an agent with a prior belief $\mu_1 \geq \mu_0$ and using the same number of samples-$n$. Observe that if $\Pr[p(n, \mu_1; \mu_b)) \leq \alpha] \geq Pr[p(n', \mu_0; \mu_b)) \leq \alpha]$, then the agent will participate under belief $\mu_1$. To compute $\Pr[p(n, \mu_1; \mu_b) \leq \alpha]$, let $n\hat{\mu} = \sum_{i=1}^{n} X_i$ denote the observed outcomes of $n$ samples drawn independently from the distribution with effectiveness $\mu_1$. Observe that since the same number of samples are used here as when the belief was $\mu_0$, the critical region $z_{\alpha,n}$ does not change since it depends only on $\alpha, n$ and $\mu_b$. The probability that these samples, drawn with respect to belief $\mu_1$, will lie in this critical region:

$$\sum_{i \in z_{\alpha,n}} \binom{n}{i} \mu_1^i (1 - \mu_1)^{n-i} \geq \sum_{i \in z_{\alpha,n}} \binom{n}{i} \mu_0^i (1 - \mu_0)^{n-i}$$

where the inequality follows immediately since $\mu_1 \geq \mu_0$.

For the reverse, consider an agent with effectiveness $\mu_0'$ not participating. This means that *for all* $n$, we have $\Pr[p(n, \mu_0'; \mu_b)) \leq \alpha] \cdot R_0 < (c \cdot n + c_0)$. Then for an agent with belief $\mu_1' < \mu_0'$ the following holds:

$$\forall n \sum_{i \in z_{\alpha,n}} \binom{n}{i} \mu_1'^i (1 - \mu_1')^{n-i} < \sum_{i \in z_{\alpha,n}} \binom{n}{i} \mu_0'^i (1 - \mu_0')^{n-i}$$

In other words, for each $n$, the passing probability under $\mu_1'$ is worse than $\mu_0'$, and the agent was already not participating under $\mu_0'$ for any $n$. $\qquad\square$

**Proof for Lemma 3.2**

*Proof.* We first consider solving for $\mu_\tau(\alpha)$, given an $\alpha$. To solve this upto some accuracy $\varepsilon$, we can discretize the belief space into $k = \frac{1}{\varepsilon}$ intervals of size $\varepsilon$, and use the mid-point of the interval as its representative belief. The monotonicity of beliefs implies that we can run a binary search over these intervals. Starting with the middle interval $\frac{k}{2}$, if the agent participates at its representative belief, we need not search intervals larger than this. Similarly, if the agent does not participate, we need not search all intervals smaller than this. Clearly, this terminates in $\log\left(\frac{1}{\varepsilon}\right)$ calls to the pass probability oracle. To check participation, we need to compute the optimal $n$ at every belief. Assume the maximum number of sample is $n_{max}$, it will again take $\log(n_{max})$ to search for the optimal number of samples. So the total complexity should be $\log\left(\frac{1}{\varepsilon}\right) * \log(n_{max})$.

Next, we show that $\mu_\tau(\alpha)$ is non-increasing as $\alpha$ increases. For any $\alpha$, we know that at the corresponding threshold belief $\mu_\tau(\alpha)$ and its corresponding optimal sample size $n_{\mu_\tau}(\alpha) \triangleq n_\tau(\alpha)$, the utility is 0. In other words: $R \cdot \text{Pass}(n_\tau(\alpha), \mu_\tau(\alpha), \mu_b, \alpha) = cn_\tau(\alpha) + c_0$. It is known that for a

fixed effect size and number of samples, the passing probability of a hypothesis test increases in $\alpha$. In other words, for two p-values $\alpha_1, \alpha_2$ where $\alpha_1 \leq \alpha_2$, the following holds:

$$\text{Pass}(\alpha_1, \mu_\tau(\alpha_1), n_\tau(\alpha_1)) \leq \text{Pass}(\alpha_2, \mu_\tau(\alpha_2), n_\tau(\alpha_2)) \quad \text{and} \quad u(n_\tau(\alpha_1), \mu_\tau(\alpha_1), \alpha_2) \geq 0$$

Since the agent with effectiveness $\mu_\tau(\alpha_1)$ will have non-negative utility when using $n_\tau(\alpha_1)$ samples when the p-value is $\alpha_2$, this agent will participate at this p-value (but not necessarily using $n_\tau(\alpha_1)$ samples as that may not be optimal). The monotonicity of participation (Lemma 3.1) implies that under $\alpha_2$, agents with effectiveness greater than $\mu_\tau(\alpha_1)$ will also participation. Thus, the participation threshold under $\alpha_2$ by definition must be to the left of $\mu_\tau(\alpha_1)$ - i.e. $\mu_\tau(\alpha_2) \leq \mu_\tau(\alpha_1)$ as desired. $\quad\square$

**Proof for Lemma 3.3**

*Proof.* We prove by contradiction. Assume that there exists an $\alpha$ and $\mu_1 \geq \mu_0$ such that $u(\alpha, \mu_0, n_{\mu_0}(\alpha)) \geq u(\alpha, \mu_1, n_{\mu_1}(\alpha))$. First note that if the agent is not participating in either, then the utility is always 0. If they participate in one and not the other, the monotonicity of participation ( Lemma 3.1) means it must be under $\mu_1$ (giving positive utility), with $\mu_b$ being 0. Thus, the only situation where the initial claim could hold if the agent participates under both. We divide this into the following three cases:

- $\mu_0 \leq \mu_1 \leq \mu_b$: since both $\mu_0$ and $\mu_1$ are less effective than the baseline drug, Theorem 3.1 implies that in both cases, $n_{min}$ samples are used, and in Lemma 3.1 we know that in such settings, for a fixed $n$, the pass probability increases in $\mu$. Thus the utility under $\mu_1$ cannot be lower than $\mu_0$.

- $\mu_0 \leq \mu_b \leq \mu_1$ and $\mu_b \leq \mu_0 \leq \mu_1$: We know from Lemma 3.1 for every fixed $n$, the pass probability in this regime increases in $\mu$. Thus, $\forall n, u(\alpha, \mu_0, n) \leq u(\alpha, \mu_1, n)$. Since $n_{\mu_1}(\alpha)$ is the optimal number of samples for $\mu_1$, we have

$$u(\alpha, \mu_0, n_{\mu_1}(\alpha)) \leq u(\alpha, \mu_1, n_{\mu_1}(\alpha)) \leq u(\alpha, \mu_0, n_{\mu_0}(\alpha))$$

The last step is according to the contradiction statement. However, this cannot be true since an $n_{\mu_1}(\alpha)$ could not be the optimal number of samples for $\mu_1$ if it led to a lower utility than $n_{\mu_0}(\alpha)$.

$\square$

## C   Omitted Proof for Section 4

For the ease of notation, let $Q$ denote the CDF of the effectiveness distribution $q$. Further, let $Q_{<\mu_b}$ and $Q_{\geq\mu_b}$ denote the CDF of the belief distribution $q$, conditioned on $\mu_0 < \mu_b$ and $\mu_0 \geq \mu_b$.

**Proof of Lemma 4.1**

*Proof.* We are interested in the total derivative of the pass probability with respect to $\alpha$ for any agent with effectiveness $\mu_0 \geq \mu_b$. Note that for this lemma, we consider the agent to always be participating. This total derivative can be expanded using the multi-variable chain rule as follows:

$$\frac{d}{d\alpha}\text{Pass}(\alpha, \mu_0, n_{\mu_0}(\alpha)) = \underbrace{\frac{\partial \text{Pass}}{\partial \alpha}}_{\text{Direct effect}} + \underbrace{\frac{\partial \text{Pass}}{\partial n} \cdot \frac{dn_{\mu_0}(\alpha)}{d\alpha}}_{\text{indirect effect}}. \tag{9}$$

To simplify notation, denote $w_\alpha = \Phi^{-1}(1 - \alpha)$, and let

$$\xi(n, \alpha) = \frac{\Phi^{-1}(1-\alpha)\sigma_b - \sqrt{n}(\mu_0 - \mu_b)}{\sigma_0} = \frac{\sigma_b}{\sigma_0}w_\alpha - \frac{\sqrt{n}(\mu_0 - \mu_b)}{\sigma_0}$$

Then we have: $\text{Pass}(\alpha, \mu_0, n_{\mu_0}(\alpha)) = 1 - \Phi(\xi(n_{\mu_0}(\alpha), \alpha))$. We now separate the analysis based on the direct and indirect influence.

**Direct effect:** The partial derivative with respect to $\alpha$ is:

$$\frac{\partial \text{Pass}}{\partial \alpha} = -\frac{d}{d\alpha}\Phi(\xi(n, a)) = -\phi(\xi) \cdot \frac{\partial \xi}{\partial \alpha} = -\phi(\xi)\frac{\sigma_b}{\sigma_0}\frac{dw_\alpha}{d\alpha} = \frac{\sigma_b}{\sigma_0}\frac{\phi(\xi)}{\phi(z_\alpha)} \geq 0$$

since a higher $\alpha$ lowers the critical threshold $w_\alpha$ and makes the test easier to pass.

**Indirect effect:** The indirect effect depends on the optimal number of samples the agent uses for a given $\alpha$. That is, while having more samples increases the chance of passing ($\frac{\partial \text{Pass}}{\partial n} > 0$), the agent might reduce this effort when $\alpha$ increases (since the test becomes easier). The product of these two terms is therefore unclear. Formally,

$$\frac{\partial \text{Pass}}{\partial n} = -\frac{d}{dn}\Phi(\xi(n,\alpha)) = -\phi(\xi)\frac{\partial \xi}{\partial n} = \phi(\xi) \cdot \frac{\mu_0 - \mu_b}{2\sigma_0\sqrt{n}} > 0$$

To compute $\frac{dn_{\mu_0}(\alpha)}{d\alpha}$, let $F(n,\alpha)$ be the first-order condition of the agent's utility function: $u(\alpha, \mu_0, n) = R\text{Pass}(\alpha, \mu_0, n) - cn - c_0$. Since we know $n_{\mu_0}(\alpha)$ is the optimal number of samples, we claim that:

$$F(\alpha, n_{\mu_0}(\alpha)) = R \cdot \frac{\partial \text{Pass}}{\partial n}\bigg|_{n=n_{\mu_0}(\alpha)} - c = 0. \tag{10}$$

This holds because, as we show immediately below, the second derivative of the $\text{Pass}(\cdot)$ function is always strictly negative with respect to $n$, meaning the utility function is always strictly concave. This is formalized below:

**Lemma C.1.** *The second derivative of Pass with respect to $n$ is always negative –* $\frac{\partial^2 \text{Pass}}{\partial n^2} < 0$.

*Proof.* To see this,

$$\frac{\partial^2 \text{Pass}}{\partial n^2} = \frac{d}{dn}\left(\phi(\xi) \cdot \frac{(\mu_0 - \mu_b)}{2\sigma_0\sqrt{n}}\right) = \phi(\xi) \cdot \left(-\frac{\xi(\mu_0 - \mu_b)^2}{4\sigma_0^2 n} - \frac{(\mu_0 - \mu_b)}{4\sigma_0 n^{3/2}}\right).$$

Since $\phi(\xi) > 0$, $\mu_0 - \mu_b > 0$, $\sigma_0 > 0$, and $n > 0$, and since $\xi$ can be either sign, the two terms in the parentheses are both negative (even if $\xi < 0$, the negative sign in front ensures negativity). Thus, the entire expression is strictly negative: $\frac{\partial^2 \text{Pass}}{\partial n^2} < 0$. Intuitively this means that every new sample increases the chance of passing, but each one helps less than the last. $\square$

The result above also means that the first derivative of $F(\alpha, n)$ is always non-zero at $n_{\mu_0}(\alpha)$. It is also evident that $F(\alpha, n)$ is continuously differentiable. Thus, we can apply the Implicit Function Theorem. Differentiating both sides of $F(\alpha, n_{\mu_0}(\alpha)) = 0$ with respect to $\alpha$ yields:

$$\frac{\partial F}{\partial n}(\alpha, n_{\mu_0}(\alpha)) \cdot \frac{dn_{\mu_0}(\alpha)}{d\alpha} + \frac{\partial F}{\partial \alpha}(\alpha, n_{\mu_0}(\alpha)) = 0 \implies \frac{dn_{\mu_0}(\alpha)}{d\alpha} = -\frac{\frac{\partial F}{\partial \alpha}}{\frac{\partial F}{\partial n}} \tag{11}$$

Next we compute $\frac{\partial F}{\partial \alpha}$ and $\frac{\partial F}{\partial n}$ accordingly. Observe that:

$$\frac{\partial F}{\partial \alpha} = R_0 \cdot \frac{\partial^2 \text{Pass}}{\partial n \partial \alpha}, \quad \text{where} \quad \frac{\partial^2 \text{Pass}}{\partial n \partial \alpha} = \frac{\mu_0 - \mu_b}{2\sigma_0\sqrt{n}} \cdot \frac{\partial \phi(\xi)}{\partial \alpha}.$$

Since:

$$\frac{\partial \phi(\xi)}{\partial \alpha} = \frac{d\phi(\xi)}{d\xi} \cdot \frac{\partial \xi}{\partial \alpha} = -\xi\phi(\xi) \cdot \frac{\partial \xi}{\partial \alpha} = \xi\phi(\xi) \cdot \frac{\sigma_b}{\sigma_0}\frac{1}{\phi(w_\alpha)},$$

we have

$$\frac{\partial F}{\partial \alpha} = R_0 \cdot \frac{\mu_0 - \mu_b}{2\sigma_0\sqrt{n}} \cdot \xi\phi(\xi) \cdot \frac{1}{\phi(z_\alpha)}\frac{\sigma_b}{\sigma_0}.$$

We next turn to computing the derivative of $F$ with respect to $n$. Observe the following (where we plug in $\frac{\partial^2 \text{Pass}}{\partial n^2}$ from above):

$$\frac{\partial F}{\partial n} = R_0 \cdot \frac{\partial^2 \text{Pass}}{\partial n^2} = R_0 \cdot \left(\frac{\xi\phi(\xi)(\mu_0 - \mu_b)^2}{4\sigma_0^2 n} - \frac{\phi(\xi)(\mu_0 - \mu_b)}{4\sigma_0 n^{3/2}}\right).$$

Now plugging them back into the expression computed by the Implicit function theorem, we have:

$$\frac{dn_{\mu_0}(\alpha)}{d\alpha} = -\frac{\frac{\partial F}{\partial \alpha}}{\frac{\partial F}{\partial n}}\bigg|_{n=n_{\mu_0}(\alpha)} = -\frac{\frac{(\mu_0 - \mu_b)\,\xi\,\phi(\xi)}{2\sigma_0\sqrt{n_{\mu_0}(\alpha)}\,\phi(w_\alpha)}\frac{\sigma_b}{\sigma_0}}{\frac{\xi\phi(\xi)(\mu_0 - \mu_b)^2}{4\sigma_0^2 n_{\mu_0}(\alpha)} - \frac{\phi(\xi)(\mu_0 - \mu_b)}{4\sigma_0 n_{\mu_0}(\alpha)^{3/2}}} = -\frac{\frac{2\xi\sqrt{n_{\mu_0}(\alpha)}\sigma_b}{\phi(w_\alpha)}}{\xi(\mu_0 - \mu_b) - \frac{\sigma_0}{\sqrt{n_{\mu_0}(\alpha)}}}.$$

**Total derivative.** Having computed the direct and indirect effects, we can directly compute the total derivative. We have:

$$\frac{d}{d\alpha}\text{Pass}(\alpha, \mu_0, n_{\mu_0}(\alpha)) = \underbrace{\frac{\partial\text{Pass}}{\partial\alpha}}_{\text{Direct effect}} + \underbrace{\frac{\partial\text{Pass}}{\partial n}\bigg|_{n=n_{\mu_0}(\alpha)} \cdot \frac{dn_{\mu_0}(\alpha)}{d\alpha}}_{\text{indirect effect}}. \tag{12}$$

$$= \frac{\sigma_b}{\sigma_0}\frac{\phi(\xi)}{\phi(z_\alpha)} + \phi(\xi) \cdot \frac{\mu_0 - \mu_b}{2\sigma_0\sqrt{n_{\mu_0}(\alpha)}} \cdot \frac{dn_{\mu_0}(\alpha)}{d\alpha} \tag{13}$$

$$= \frac{\sigma_b}{\sigma_0}\frac{\phi(\xi)}{\phi(z_\alpha)}\left(1 - \frac{(\mu_0 - \mu_b)\,\xi}{\left(\xi(\mu_0 - \mu_b) - \frac{\sigma_0}{\sqrt{n_{\mu_0}(\alpha)}}\right)}\right). \tag{14}$$

The sign of $\left(1 - \frac{(\mu_0 - \mu_b)\,\xi}{\left(\xi(\mu_0 - \mu_b) - \frac{\sigma_0}{\sqrt{n_{\mu_0}(\alpha)}}\right)}\right)$ decides the monotonicity of the passing probability. As we show below, when agent chooses the optimal number of samples $n = n_{\mu_0}(\alpha)$, it will never be the case that $\xi(\mu_0 - \mu_b) - \frac{\sigma_0}{\sqrt{n(\alpha)}} > 0$ (Lemma C.2), which implies that $\left(1 - \frac{(\mu_0 - \mu_b)\,\xi}{\left(\xi(\mu_0 - \mu_b) - \frac{\sigma_0}{\sqrt{n_{\mu_0}(\alpha)}}\right)}\right) > 0$ always holds at the optimal $n_{\mu_0}(\alpha)$, thus the passing probability at the optimal number of samples $n_{\mu_0}(\alpha)$ is always monotonically increasing. We finish the proof by proving Lemma C.2:

**Lemma C.2.** *When $n_{\mu_0}(\alpha)$ is the optimal sample size chosen by the agent to maximize utility, then $\xi(\mu_0 - \mu_b) - \frac{\sigma_0}{\sqrt{n(\alpha)}} > 0$ never holds.*

*Proof.* Recall the second derivative of the utility w.r.t $n$ is:

$$\frac{\partial^2\text{Utility}(n, \mu_0; \mu_b, \alpha)}{\partial n^2} = \frac{\partial^2\text{Pass}}{\partial n^2}$$

$$= \frac{\xi\phi(\xi)(\mu_0 - \mu_b)^2}{4\sigma_0^2 n} - \frac{\phi(\xi)(\mu_0 - \mu_b)}{4\sigma_0 n^{3/2}}$$

$$= \frac{\phi(\xi)(\mu_0 - \mu_b)}{4\sigma_0^2 n}(\xi(\mu_0 - \mu_b) - \frac{\sigma_0}{\sqrt{n}})$$

At the optimal number of sample $n = n_{\mu_0}(\alpha)$, the second derivative of the utility must be negative, otherwise it implies that increasing the number of samples will increase the utility, which leads to a contradiction to the fact that $n = n_{\mu_0}(\alpha)$ is an optimal number of samples, thus we have the following always holds at $n = n_{\mu_0}(\alpha)$: $\xi(\mu_0 - \mu_b) - \frac{\sigma_0}{\sqrt{n_{\mu_0}(\alpha)}} \leq 0$. □

□

**Proof of Theorem 4.2**

*Proof.* Recall that the false negative rate in our setting consists of two components, conditioned on whether agents participate. Observing that when an agent does not participate – i.e. $\mu_0 < \mu_\tau(\alpha)$ – their probability of passing the statistical test is 0, we can simplify the overall FN rate as follows:

$$\text{FN}(\alpha, \mathcal{I}) = \underset{\mu_0 \sim q}{\mathbb{E}}\left[\text{Fail}(\alpha, \mu_0, n_{\mu_0}(\alpha))|\mu_0 \geq \mu_b\right]$$

$$= \underbrace{\underset{\mu_0 \sim q}{\mathbb{E}}\left[\text{Fail}(\alpha, \mu_0, n_{\mu_0}(\alpha))|\mu_0 \geq \mu_b, \mu_0 \geq \mu_\tau(\alpha)\right]P(\mu_0 \geq \mu_\tau(\alpha)|\mu_0 \geq \mu_b)}_{\text{FN}_{\text{particip}}}$$

$$+ \underbrace{P[\mu_0 \leq \mu_\tau(\alpha)|\mu_0 \geq \mu_b]}_{\text{FN}_{\text{abstain}}}$$

Consider first any $\alpha_1 \leq \widehat{\alpha}$. We can explicitly express each component of the false negative as follows, since we are guaranteed that $\mu_\tau(\alpha_1) \geq \mu_b$:

$$\text{FN}_{\text{particip}}(\alpha_1, \mathcal{I}) = \frac{1 - Q(\mu_\tau(\alpha_1))}{1 - Q(\mu_b)} \int_{\mu_\tau(\alpha_1)}^1 \text{Fail}(\alpha_1, \mu_0, n_{\mu_0}(\alpha)) \frac{q(\mu_0)}{1 - Q(\mu_\tau(\alpha_1))} d\mu_0 \quad (15)$$

$$\text{FN}_{\text{abstain}}(\alpha_1, \mathcal{I}) = \frac{Q(\mu_\tau(\alpha_1)) - Q(\mu_b)}{1 - Q(\mu_b)} \quad (16)$$

Let us focus on $\text{FN}_{\text{particip}}$. To simplify this, the following result is helpful. For a positive function $g(x)$ and a function $f(x)$ with minimum and maximum values $f_{min}$ and $f_{max}$ over an interval $[a, b]$, the following holds:

$$f_{min} \int_a^b g(x) \leq \int_a^b g(x)f(x)dx \leq f_{max} \int_a^b g(x) \implies f_{min} \leq \frac{\int_a^b g(x)f(x)dx}{\int_a^b g(x)} \leq f_{max}$$

Since the expression in the middle is between $f_{min}$ and and $f_{max}$, by the intermediate value theorem, there exists an $c \in [a, b]$ such that: $f(c) = \frac{\int_a^b g(x)f(x)dx}{\int_a^b g(x)}$ which means: there exists a $c$ such that $f(c) \int_a^b g(x) = \int_a^b f(x)g(x)$. This can be interpreted as an integral mean value theorem. Using this, there exists a value $\mu_c^1 \in [\mu_\tau(\alpha_1), 1]$ such that the $\text{FN}_{particip}(\alpha_1)$ can be expressed as follows:

$$\text{FN}_{\text{particip}}(\alpha_1, \mathcal{I}) = \frac{1 - Q(\mu_\tau(\alpha_1))}{1 - Q(\mu_b)} \text{Fail}(\alpha_1, \mu_c^1, n_{\mu_c^1}(\alpha)) \int_{\mu_\tau(\alpha_1)}^1 \frac{q(\mu_0)}{1 - Q(\mu_\tau(\alpha_1))} d\mu_0$$

$$= \frac{1}{1 - Q(\mu_b)} \text{Fail}(\alpha_1, \mu_c^1, n_{\mu_c^1}(\alpha)) \int_{\mu_\tau(\alpha_1)}^1 \frac{q(\mu_0)}{1 - Q(\mu_\tau(\alpha_1))} d\mu_0$$

$$= \frac{1 - Q(\mu_\tau(\alpha_1))}{1 - Q(\mu_b)} \text{Fail}(\alpha_1, \mu_c^1, n_{\mu_c^1}(\alpha))$$

Therefore, we can write:

for some $\mu_c^1 \in [\mu_\tau(\alpha_1), 1]$: $\text{FN}(\alpha_1, \mathcal{I}) = \frac{1 - Q(\mu_\tau(\alpha_1))}{1 - Q(\mu_b)} \text{Fail}(\alpha_1, \mu_c^1, n_{\mu_c^1}(\alpha)) + \frac{Q(\mu_\tau(\alpha_1)) - Q(\mu_b)}{1 - Q(\mu_b)}$

Now consider an $\alpha_2$ such that $\alpha_1 < \alpha_2 \leq \alpha^*$. Note that due to Appendix B, $\mu_\tau(\alpha_2) \leq \mu_\tau(\alpha_1)$. We can thus write the $\text{FN}_{\text{particip}}$ of this instance as follows:

$$\text{FN}_{\text{particip}}(\alpha_2, \mathcal{I}) = \frac{1 - Q(\mu_\tau(\alpha_2))}{1 - Q(\mu_b)} \int_{\mu_\tau(\alpha_2)}^1 \text{Fail}(\alpha_2, \mu_0, n_{\mu_0}(\alpha)) \frac{q(\mu_0)}{1 - Q(\mu_\tau(\alpha_2))} d\mu_0$$

$$= \frac{1}{1 - Q(\mu_b)} \left[ \int_{\mu_\tau(\alpha_2)}^{\mu_\tau(\alpha_1)} \text{Fail}(\alpha_2, \mu_0, n_{\mu_0}(\alpha_2))q(\mu_0)d\mu_0 + \int_{\mu_\tau(\alpha_1)}^1 \text{Fail}(\alpha_2, \mu_0, n_{\mu_0}(\alpha_2))q(\mu_0)d\mu_0 \right]$$

$$= \frac{1}{1 - Q(\mu_b)} \left[ \text{Fail}(\alpha_2, \mu_c^{2,1}, n_{\mu_c^{2,1}}(\alpha)) \int_{\mu_\tau(\alpha_2)}^{\mu_\tau(\alpha_1)} q(\mu_0)d\mu_0 + \text{Fail}(\alpha_2, \mu_c^{2,2}, n_{\mu_c^{2,2}}(\alpha)) \int_{\mu_\tau(\alpha_1)}^1 q(\mu_0)d\mu_0 \right]$$

$$= \text{Fail}(\alpha_2, \mu_c^{2,1}, n_{\mu_c^{2,1}}(\alpha)) \frac{Q(\mu_\tau(\alpha_1)) - Q(\mu_\tau(\alpha_2))}{1 - Q(\mu_b)} + \text{Fail}(\alpha_2, \mu_c^{2,2}, n_{\mu_c^{2,2}}(\alpha)) \frac{1 - Q(\mu_\tau(\alpha_1))}{1 - Q(\mu_b)}$$

where in the third transition, we use the integral mean value theorem as in the previous $\alpha_1$ case. Note that for the integral between $[\mu_\tau(\alpha_1), 1]$ in the $\alpha_2$ setting, we know that for every $\mu_0 \in [\mu_\tau(\alpha_1), 1]$, $\text{Fail}(\alpha_2, \mu) < \text{Fail}(\alpha_1, \mu)$ since from Lemma 4.1, we know that the pass probability increases as alpha increases and this set of agents participated under both $\alpha_1$ and $\alpha_2$. This immediately means that $\text{Fail}(\alpha_2, \mu_c^{2,2}, n_{\mu_c^{2,2}}(\alpha)) \leq \text{Fail}(\alpha_2, \mu_c^1, n_{\mu_c^{2,1}}(\alpha))$. In the $[\mu_\tau(\alpha_2), \mu_\tau(\alpha_1)]$, the failure probability is at most 1. Hence, $\text{Fail}(\alpha_2, \mu_c^{2,1}, n_{\mu_c^{2,1}}(\alpha)) \leq 1$. Thus, we can upper bound the FN participation loss at $\alpha_2$ as follows:

$$\text{FN}_{\text{particip}}(\alpha_2) \leq \frac{Q(\mu_\tau(\alpha_1)) - Q(\mu_\tau(\alpha_2))}{1 - Q(\mu_b)} + \text{Fail}(\alpha_1, \mu_c^1, n_{\mu_c^1}(\alpha)) \frac{1 - Q(\mu_\tau(\alpha_1))}{1 - Q(\mu_b)} \quad (17)$$

We now express the abstain loss for $\alpha_2$ as follows:

$$\text{FN}_{\text{abstain}}(\alpha_2) = \frac{Q(\mu_\tau(\alpha_2)) - Q(\mu_b)}{1 - Q(\mu_b)} = \frac{Q(\mu_\tau(\alpha_1)) - Q(\mu_b)}{1 - Q(\mu_b)} - \frac{Q(\mu_\tau(\alpha_1)) - Q(\mu_\tau(\alpha_2))}{1 - Q(\mu_b)}$$

Then we have that:

$$\text{FN}(\alpha_2, \mathcal{I}) = \text{FN}_{\text{particip}} + \text{FN}_{\text{abstain}} \tag{18}$$

$$\leq \text{Fail}(\alpha_1, \mu_c^1, n_{\mu_c^1}(\alpha)) \frac{1 - Q(\mu_\tau(\alpha_1))}{1 - Q(\mu_b)} + \frac{Q(\mu_\tau(\alpha_1)) - Q(\mu_b)}{1 - Q(\mu_b)} = \text{FN}(\alpha_1, \mathcal{I}) \tag{19}$$

Lastly, for any $\alpha \geq \hat{\alpha}$, we know from Proposition 4.2 that $\text{FN}_{\text{particip}}(\alpha, I) = 0$ and from Proposition 4.1 that $\text{FN}_{\text{abstain}}(\alpha, \mathcal{I}) = 0$. $\qquad \square$

# D Discussion on Sensitivity Analysis

In this section, we discuss how our framework can be naturally extended to accommodate small misspecifications on either the agent or principal side, and why our main insights remain stable under such uncertainty.

Both the agent's participation decision and the principal's optimal p-value threshold $\hat{\alpha}$ depend only on the cost and revenue parameters $(c_0, c, R)$. In realistic settings such as drug development, these parameters are estimated from historical data or business forecasts and may contain some error. We can therefore model parameter uncertainty by assuming that the agent and principal each have slightly perturbed estimates of the true values. In particular, we assume the agent has access to approximate parameters $(\hat{c}_0, \hat{c}, \hat{R})$ within $\epsilon$ of the true values, and let the principal have their own estimates $(\tilde{c}_0, \tilde{c}, \tilde{R})$ within $\delta$ of the true values. The magnitudes of $\epsilon$ and $\delta$ can differ across parameters.

**Agent-side uncertainty.** We first examine how such $\epsilon$-level errors affect the agent's decision for a given belief $\mu_0$. Extending Theorem 3.1, recall that the utility function $u(n; \alpha)$ can be partitioned by indices $(n_1, n_2)$ where it is either convex or concave. As noted in Equation (8) of Appendix B, these indices are invariant to small parameter perturbations. In the convex region, the optimal $n$ occurs at an endpoint and is therefore unchanged. In the concave region, the optimum corresponds to the $x$-intercept of $\partial u / \partial n$. A perturbation of size $\epsilon$ shifts this derivative by $O(\epsilon)$, implying that the optimal $n$ moves by at most $O(\epsilon)/L$, where $L$ is the inverse Lipschitz constant of $\partial u / \partial n$[6]:

$$|\hat{n}_{\mu_0}(\alpha) - n_{\mu_0}(\alpha)| \leq O(\epsilon)/L.$$

A similar argument bounds the difference in participation thresholds:

$$|\hat{\mu}_\tau(\alpha) - \mu_\tau(\alpha)| \leq O(\epsilon)/k,$$

where $k$ is the inverse Lipschitz constant (with respect to $\mu_0$) of $u(\mu_0, n(\alpha, \mu_0))$.

**Principal-side uncertainty.** If the principal's perceived parameters $(\tilde{c}_0, \tilde{c}, \tilde{R})$ differ from those of the agent, they will infer a different threshold belief $\tilde{\mu}_\tau(\alpha)$. Without observing the agent's $\epsilon$, the principal optimizes $\alpha$ under their own model, obtaining $\hat{\alpha}$ such that $\tilde{\mu}_\tau(\hat{\alpha}) = \mu_b$. However, because $\epsilon \neq \delta$, the agent's actual threshold $\hat{\mu}_\tau(\hat{\alpha})$ may not equal $\mu_b$. This deviation can again be bounded by the inverse Lipschitz constant $k$:

$$|\hat{\mu}_\tau(\hat{\alpha}) - \mu_b| \leq O(|\epsilon - \delta|)/k.$$

The agent's deviation depends only on the difference between their perceived parameters and the true ones, whereas the principal's deviation depends on the gap between their perception and the agent's. These uncertainties can thus be analyzed independently. Overall, small estimation errors in cost or revenue induce proportionally small and bounded shifts in the optimal participation and threshold decisions.

---

[6]by inverse lipschitz constant, we mean $|f(x_1) - f(x_2)| \geq L|x_1 - x_2|$, allowing us to bound change in the domain given change in range

**Stability of our results** Building on the above observations, we now explain why the main insights of our model remain stable under such uncertainty. This robustness follows from two structural properties of the model: *monotonicity* and *Lipschitz continuity*. First, both the participation threshold $\mu_\tau(\alpha)$ and the principal's optimal choice $\hat{\alpha}$ are continuous and monotone functions of the cost and revenue parameters $(c_0, c, R)$. Second, their derivatives with respect to these parameters are bounded by Lipschitz constants $k$ and $L$, ensuring that small perturbations induce at most proportional changes:

$$|\Delta\mu_\tau(\alpha)| \le k \left\| (\Delta c_0, \Delta c, \Delta R) \right\|, \quad |\Delta\hat{\alpha}| \le L \left\| (\Delta c_0, \Delta c, \Delta R) \right\|.$$

These bounds imply that the relative ordering of participation and approval thresholds is preserved: The curves may shift, but they do not cross or invert. As a result, the qualitative behavior of the model—for instance, that higher costs discourage participation and looser $\alpha$ increases it—remains unchanged.

## E   Plot for Vaccine Drugs

Figure 4: $\widehat{\alpha}$ vs Revenue - Vaccines

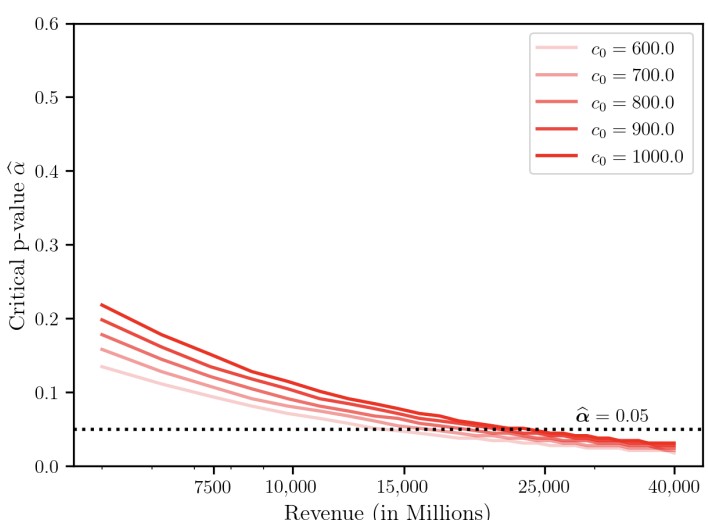

