# OpenReview forum: "Strategic Hypothesis Testing"
_NeurIPS.cc/2025/Conference — NeurIPS 2025 spotlight_

### Official Review · Reviewer_87c8 · 2025-06-24

**Clarity:** 4
**Significance:** 3
**Originality:** 3
**Rating:** 5
**Confidence:** 4

**Summary:**

Summary:

This submission develops a game-theoretic framework for hypothesis testing in a regulatory environment where an agent (who has private information about a product’s effectiveness) strategically responds to a p-value threshold set by a principal (e.g. a regulatory body like the FDA). This submission extends prior work in this direction by extending the analysis beyond binary notions of effectiveness, and allowing the agents to choose trial sample sizes. The principal’s goal is to minimize some combination of Type I and Type II errors while accounting for the agent’s strategic behavior.

This strategic interaction is modeled as a Stackelberg game, where the principal commits to a p-value, and the agent best-responds by deciding (1) whether to participate in the trial and (2) how many samples to collect if they choose to participate. In their analysis, the authors show that error rates behave monotonically, which allows them to compute the optimal p-value threshold given this strategic behavior.

Empirically, the authors apply their methods to drug approval data in oncology, cardiovascular, and vaccine settings by using publicly available data to estimate trial costs and revenue. They find that the FDA’s common p-value threshold of 0.05 closely aligns with the critical threshold for median-cost drugs. They also highlight how overly conservative thresholds may disproportionately discourage participation from low-revenue but effective drugs.

**Questions:**

How sensitive is your model to unknown revenue and cost parameters from (1) the principal’s side and (2) the agent’s side?

**Ethical Concerns:**

["NO or VERY MINOR ethics concerns only"]

**Final Justification:**

After the rebuttal, my opinion on the paper remains positive. The theoretical model and policy implications are interesting, and my concern about knowledge of model parameters has been addressed.

**Limitations:**

Yes

**Quality:**

4

**Strengths And Weaknesses:**

Strengths:

The theoretical model is well-motivated by practice, since in reality agents (1) have a strong incentive to strategize and (2) have the ability to select their own sample size. The monotonicity properties are technically interesting, with proof sketches in the main body and full proofs in the appendices. More broadly, the paper is clearly written and well-structured. The experimental results are also quite interesting, with potential real-world policy implications, which is nice as it is often difficult to run meaningful experiments outside of simulation for these types of strategic learning models.

Weaknesses:

The authors’ model assumes that the principal has perfect knowledge of the agent’s cost and revenue parameters. While this assumption may be reasonable for after-the-fact analyses like the ones presented in the experimental results of this paper, the agents themselves may not even know these parameters before the trial is run and the product is sold, even putting aside the assumption that the principal knows these parameters. It would have been nice to see some discussion around this, either justifying these assumptions or providing some sort of sensitivity analysis to the equilibrium if the players have estimates of these parameters.

---

> ### Author Rebuttal · Authors · 2025-07-30
>
> Thank you for your review and insightful comments. We are glad that you found the problem interesting and well-motivated, and the results technically interesting. We hope our work can lead to further empirical and technical works exploring strategic behaviour for hypothesis testing. Below, we include a detailed response to your question:
>
> ---
>
> > **Uncertainties about cost and revenue parameters:**
>
> Indeed, both the agent participation decision and the principal’s optimal p-value lower-bound $\hat{\alpha}$ depend on the cost and revenue parameters; in fact, these are the only parameters required for this decision. In drug development, past data on trials of similar drugs/genres can be used to estimate costs, whereas projected revenue must be computed for any business decision prior to development and trials. That said, you are correct that in practice, these parameters can only be calculated approximately/with some error. Below is a way to extend our work to such scenarios.
>
> In the most general case, we can consider the agent having access to parameters $\hat{c_0}, \hat{c}, \hat{R}$ that are within $\epsilon$ of the true values (note that epsilon can be different for each of the three; this is just for ease of exposition); similarly, the principal has access to $\tilde{c_0}, \tilde{c}, \tilde{R}$ that are within $\delta$ of the true values.
>
> **1. Agent-side approximation error:**
> We can first ask how different (due to $\epsilon$ error) the agent's decision would be upon observing the effectiveness $\mu_0$. In generalizing Theorem 3.1, we note that the analysis depends on the properties of $u(n; \alpha)$ and that it can be partitioned by indices $n_1, n_2$ such that within each partition, the function is convex or concave. As we see in equation 8 in Appendix B, the indices $n_1, n_2$ are not affected by $\epsilon$; neither is the decision in the convex region since the minimum is at the endpoint. For the concave region, the optimal corresponds to the x-intercept of $\partial u/\partial n$. The error $\epsilon$ will change the value of the utility function derivative by $O(\epsilon)$; thus the change in x-intercept (and thus the optimal choice of $n$) will shift by at most $O(\epsilon)/L$, where $L$ is the inverse Lipschitz constant of $\partial u/\partial n$ (by inverse lipschitz constant, we mean $|f(x_1) - f(x_2)| \geq L|x_1 - x_2|$, allowing us to bound change in the domain given change in range). That is, $|\hat{n} (\alpha; \mu_0) - n(\alpha; \mu_0)| \leq \frac{O(\epsilon)}{L}$. For the function $\partial u/\partial n$ specifically, this inverse lipschitz constant can be expressed in terms of  $n_1, n_2, n_{min}, n_{max}$.
>
> Uncertainty about cost and revenue also affects the participation threshold $\mu_\tau(\alpha)$, defined as the minimum belief where an agent will participate for a given p-value $\alpha$. Again, since the utility deviates by $O(\epsilon)$, the corresponding change in threshold belief can again be bounded by the inverse lipschitz constant $k$ (in terms of $\mu_0$) of the function $u(\mu_0, n(\alpha, \mu_0))$. In short, this approach can bound $\mu_{\tau}(\alpha)$ and $\hat{\mu}_{\tau}(\alpha)$.
>
> **2. Principal side approximation error:**
> As for the principal, their perception of the cost/revenue, $\tilde{c_0}, \tilde{c}, \tilde{R}$, is different than the agents. So for any $\alpha$, they will perceive a threshold belief $\tilde{\mu}_{\tau}(\alpha)$. As the principal does not know the agent's error $\epsilon$, they can do no better than follow the conclusions of section 4 and choose a p-value of at least
> $\hat{\alpha}$ such that
>
> $\tilde{\mu}_{\tau}(\hat{\alpha}) = \mu_b.$
>
> But under this $\hat{\alpha}$, since the agent's estimated parameter is different from the principal's estimated parameter ($\epsilon \neq \delta$), we know that $\hat{\mu}_{\tau}(\hat{\alpha}) \neq \mu_b$, which means that the agent will not behave exactly as the principal expects.
>
> However, it is possible to compute how far away $\hat{\mu}_{\tau}(\hat{\alpha})$ is from $\mu_b$ by using the inverse Lipschitz constant $k$ of function $u(\mu_0, n(\alpha, \mu_0))$ mentioned above.
>
> Importantly, note that the principal's uncertainty error depends only on how far their cost/revenue estimate $\tilde{c_0}, \tilde{c}, \tilde{R}$  was from the agent's estimate $\hat{c_0}, \hat{c}, \hat{R}$ (and not the true value), since this is what the agent uses to make their decision. The agent's uncertainty error, on the other hand, only depends on how far their estimate $\hat{c_0}, \hat{c}, \hat{R}$ was from the true values $c_0, c, R$. One can analyze them separately.

---

> > ### Comment · Reviewer_87c8 · 2025-08-02
> >
> > Thanks for your reply. I will maintain my score, and will engage with the other reviewers throughout the rest of the discussion period. I'd encourage the authors to include (a fleshed out version of) the above discussion about unknown parameters in the final version.

---

> > > ### Author Response · Authors · 2025-08-04
> > >
> > > Thank you for your response. We will indeed formalize the argument about uncertain parameters in the final version.

---

### Official Review · Reviewer_vgRW · 2025-07-02

**Clarity:** 4
**Significance:** 2
**Originality:** 3
**Rating:** 4
**Confidence:** 4

**Summary:**

The paper studies a principal-agent setting applied to hypothesis testing. The goal of the principal is to set a p-value in order to control the rate of false positive and false negative. The paper characterizes how the agent behavior changes depending on the p-value. Moreover, the authors characterize the dependence of the errors on the set p-value. The main contribution of the paper is the definition of a critical threshold $\hat \alpha$ such that the agent's error is monotonic for both $\alpha<\hat \alpha$ and $\alpha>\hat \alpha$. This implies that setting the $p$-value to $\hat \alpha$ is a good choice in practice. Finally, the theoretical claims are validated by an empirical analysis.

**Questions:**

Do you have any insight on the structure of the optimal p-value in more general settings?

**Ethical Concerns:**

["NO or VERY MINOR ethics concerns only"]

**Final Justification:**

The authors adressed my questions. My evaluation of the paper is unchanged.

**Limitations:**

None.

**Paper Formatting Concerns:**

None.

**Quality:**

3

**Strengths And Weaknesses:**

Strengths:
The problem is interesting and well-motivated. Moreover, the paper is well-written and easy to follow. Finally, the paper provides a clear characterization of the problem.

Weaknesses:
The claim that the paper characterizes the optimal $p$-value is false, at least for the general problem introduced in the preliminaries (line 153). The only theoretical result is that the optimal value is larger than $\hat \alpha$. This is (approximately) optimal only when $\lambda_{f_p}>>\lambda_{f_n}$. It would be great to study also the general problem or change the focus of the paper to a more restrict scenario in which false positive should be avoided.

The technical contribution is quite weak, and limited to some calculations to characterize whether the rates of error increase or decrease with the increase of the  $p$-value.




Minor:

Line 158, a a

Line 192, to for those with

Line 198, missing space

Line 222, $\lambda_1$ non consistent with the notation.

Line 296, Theorem []

Line 300, the statement of the theorem is a bit confusing, the function is always non increasing
Line 982, $\alpha^*-> \hat \alpha$?

---

> ### Author Rebuttal · Authors · 2025-07-30
>
> Thank you for your thoughtful review and comments. We are glad you found the work well-motivated and interesting. Indeed, given the breadth and scope of hypothesis testing and statistical decision-making in the wild (e.g., food, drugs, manufacturing), we hope this work opens up a broader research direction on designing optimal decision rules that account for agent incentives and strategic manipulation. Please see below detailed answers to questions/comments in the review:
>
> ---
>
> > **Regarding the optimal p-value:**
>
> You are right that the value $\hat{\alpha}$ we rigorously characterize is not the global optimum for the general problem; rather, it is a lower bound on the optimal p-value threshold $\hat{\alpha} \: \alpha^* \geq \hat{\alpha}$.
> 1. Practicality: In most settings (like drug approvals), the natural desire is to choose a low p-value as this bounds the false positive rate. As a result, bounding the optimal p-value from below may be sufficient in practice, as regulators/principals are generally adverse to higher p-values. Moreover, since our problem is always constant-dimensional (effectiveness is a scalar), if computing the exact $\alpha^*$ is desired, this can be easily done empirically, by plotting the cumulative loss and noting the minimum (see Figure 1 for example). We include code for this computation.
> 2. Interpretability: As our analysis shows, $\hat{\alpha}$ dominates any p-value below it in the sense that both FN and FP are monotonically worse for such values. Above $\hat{\alpha}$, the relative strengths of FN and FP dictate this trade-off. Since the value of $\hat{\alpha}$ is agnostic to both the underlying distribution $q$ and the hyper-parameters $\lambda_{fn}$ and $\lambda_{fp}$ (parameters which may be noisy/heuristic driven), this is a much easier decision to justify. Note that the global optimal $\alpha^*$ will depend on all these parameters – a formal analysis of this would compare the derivatives $\frac{d}{d\alpha} \text{FN}(\alpha)$ and $\frac{d}{d\alpha} \text{FP}(\alpha)$ which depend on parameters (q, $\lambda_{fn}$, $\lambda_{fp}$).
>
> In the revised version, we will highlight these operational advantages and interpretability of $\hat{\alpha}$ and comment on the properties and comparisons to $\alpha^*$.
>
> ---
>
> > **Regarding the technical contribution:**
>
> Our goal at the onset was to propose a realistic and principled model of strategic interaction in the hypothesis testing framework. Obtaining clear and interpretable conclusions seemed difficult here; unlike standard settings, the principal must control both statistical error and agent participation using only the p-value threshold $\alpha$. The challenge is further compounded by the rich interplay among the prior, passing probabilities, sample sizes, and agents’ strategic behavior. Some of these relations cannot be explicitly characterized and are not smooth. Yet a sequence of subtle observations and technical insights results in a set of clean structural and practically relevant results. The clarity and easy interpretability of such results are, in our view, a key strength of the framework, especially since decision-makers in these contexts, such as regulators, must justify their choices to a broad audience.
>
> Moreover, while the results are interpretable, the conclusions can be surprising. The fact that a lower bound on the optimal p-value emerges from our work is distinct from the classic thinking of upper-bounding $\alpha$ to control Type I errors. Our analysis is also flexible, highlighting that if context-dependent p-values are unpalatable, regulators can instead subsidize the fixed and marginal costs to achieve similar results. We will highlight the latter point more in the revision.

---

> > ### Comment · Reviewer_vgRW · 2025-08-05
> >
> > Thanks for the response. I will keep my positive evaluation of the work.

---

### Official Review · Reviewer_uvQJ · 2025-07-03

**Clarity:** 3
**Significance:** 3
**Originality:** 3
**Rating:** 4
**Confidence:** 3

**Summary:**

The paper studies the problem of principal-agent hypothesis testing, expanding upon the framework proposed by Bates et al. (2023). More specifically, they extend the framework to include considerations of per-sample cost and thus model the agents’ (i.e. pharmaceutical firms) ability to select trial sizes in response to the principal’s (e.g. the FDA) declared significance level $\alpha$. Additionally, they incorporate false negative considerations in the principal’s loss (which is a combination of the expected incidence of false positives and of false negatives).

By analyzing the Stackleberg game arising out of the agent maximizing their utility function in response to the published significance level and the corresponding loss landscape achievable by different settings of $\alpha$, they are able to identify a critical threshold $\widehat{\alpha}$ defined as the level for which the participation threshold is exactly the baseline effectiveness. They show that for $\alpha \leq \widehat{\alpha}$: (i) the overall false positive loss is $0$ and (ii) the overall false negative loss is non-increasing, and for $\alpha > \widehat{\alpha}$: (i) the overall false positive loss is non-decreasing and (ii) the overall false negative rate is non-increasing. This suggests that the true optimal significance level $\alpha^\star$ is \emph{at least} $\widehat{\alpha}$ -- therefore using the critical value enables lower loss than a predefined threshold such as $0.05$ and the loss in optimality relative to $\alpha^\star$ is due to over-conservativeness rather than allowing more false positives.

Finally, they complement their theoretical insights with experimental studies where they compute the critical value $\widehat{\alpha}$ as a function of the revenue $R$, fixed trial cost $c_0$, and per-sample cost $c$ -- where the ranges for these variables are extracted from actual real trial data. Their experiments show that the critical value for lower revenue interventions is actually higher than the FDA imposed 0.05 (and thus the optimal $\alpha^\star$ as well and therefore simply shifting towards $\widehat{\alpha}$ would already be an improvement \emph{under the framework’s assumptions}).

**Questions:**

My main question is how do you think your framework should impact actual practice and regulatory policy?

For example, with respect to my first note in the “Limitation” section below, how seriously should the takeaway that “if the regulatory bodies wish for decreased drug prices with fixed costs being immutable, they ought to specify a higher p-value threshold” be interpreted?

And with respect to my second note, how do you think of the meaning the results in situations where $q$ and $\mu_0$ are actually not known? For $\mu_0$ could it be changed to be in terms of “belief” and could the “belief” be a bit more realistic such as uniform over a range? (and is there anything similar that could make the assumption on knowledge of $q$ more reasonable?)

Additionally, what is additional work (if any) that should be done to ensure such a framework can be actually relevant to regulatory bodies? Finally, do you have any thoughts on experiments or tests that could be done that could assess welfare effects of such theory-based policy suggestions?

**Ethical Concerns:**

["NO or VERY MINOR ethics concerns only"]

**Final Justification:**

My recommendation is for acceptance. The reason for giving a 4 rather than a 5 is based upon the descriptions of the score which state that a 5 should have "high impact on one sub-area or moderate-to-high impact on multiple areas," and to me the paper is a technically strong paper that has moderate impact on one sub-area.

**Limitations:**

I would expand the discussion on the take-away that a stricter $\alpha$ might dissuade effective but low-revenue drugs from participating without significantly increasing the possibility of false positives for very high revenue drugs. As is, the possible limitations of the framework and how the resulting take-away might not materialize is not really discussed.

Given that if the framework is to be taken seriously in a policy setting (which I am guessing would be the ideal goal, even if hard to achieve) this take-away is a very striking one that would change the outcomes of the current process significantly, I believe that a more nuanced discussion (with more pointed discussion and even ideally relevant related FDA policy papers discussing their current choice) is warranted.

Another limitation that would warrant further discussion is the assumption that the distribution $q$ is known to the principal and the agent and that the agent knows the mean effectiveness $\mu_0$. It would be good to acknowledge this is not realistic and how this affects the applicability of the results in real contexts.

**Quality:**

3

**Strengths And Weaknesses:**

A key strength is that the proposed extension of allowing agents to strategically choose their sample size in response to the principal’s announced significance level is very well aligned with practice, both in aim and in the concrete way this is added to the model. The incorporation of a false negative loss component is also appealing, though here I would say that it would be more realistic for this to scale with the effect size (for example the FDA cares less about false negatives of very marginally effective drugs than of highly effective drugs).

I also really appreciate the empirical analysis as it is grounded in actual data with appropriate citations to plug in ranges for the revenue $R$, the fixed cost $c_0$ and the per-sample costs $c$. Finally, I think it works well that proof sketches are included and that the take-aways are well-presented and it is easy to follow the logic and the results.

A potential weakness is that the distribution $q$ is assumed to be known to both the principal and the agent, and, realistically, I think neither would have even coarse knowledge of it. Likewise (though possibly less problematically), I don’t think the agents really would have knowledge of the mean effectiveness of the proposed compound $\mu_0$.

Finally, some small typos/writing considerations:
- On line 7, I think it’s meant to be “requested”
- On line 27, I am not sure what is meant by “high reputation”
- On line 139 I think it should be “the principal”
- On line 139, the “loss” was not really defined or discussed previously to an extent comparable to the utility so it is a bit abstract/hazy to say that the principal is minimizing it
- On line 156, it should be just “principal” not “principal’s”
- On line 296, there is no Theorem reference – it appears as “[]”
-	The first ‘sentence’ in Theorem 4.3 is not really a sentence
-	In Figure 1 in the RHS plot I think it was meant to be “TruncNorm”

---

> ### Author Rebuttal · Authors · 2025-07-30
>
> Thank you for your detailed review and comments, which gave us new insights into our contributions and setting! We will fix all typos in the revision; please see below a discussion on the more specific points raised:
>
> ---
>
> > **Noisy and imperfect observations of distribution $q$ and effectiveness $\mu_0$**
>
> We presently assume that the effectiveness of a product (drug) $\mu_0 \sim q$, where $\mu_0$ is the privately observed effectiveness and $q$ is the originating distribution. $q$ can be seen as the historical performance of drugs in a given category, data that the FDA typically maintains. As for $\mu_0$, we interpret this as the evidence the manufacturer has based on internal testing and trials, which typically precede a public trial. You are absolutely right that there can be noise and uncertainties about these parameters in practice. However, we note a key strength of our work/model: **neither the agent participation decision nor the p-value threshold $\hat{\alpha}$ requires access to q**. Specifically,
> 1. Knowledge of q is not required by the agent: the agent observes $\mu_0 \sim q$, and then reacts (decides to participate and chooses samples if so) based on this realization.
> 2. Our core result, the optimal p-value lower bound $\hat{\alpha}$, is agnostic to the distribution $q$. This is of major operational importance since it means the principal also does not need to know $q$ to choose $\hat{\alpha}$. That said, $q$ does affect the loss incurred by the principal at this chosen p-value. We detail this below.
>
> > **Quantify the impact of uncertainty due to estimation errors in $q$ on loss calculations**
>
> Computing the FN/FP loss at any chosen p-value, including $\hat{\alpha}$, requires access to $q$. Suppose the principal has an erroneous version of the distribution $q$ called $q_{\text{approx}}$. Then for any $\alpha$, we can bound the difference between the true FN/FP as calculated under $q$, and the approximate one calculated under $q_\text{approx}$ using TV distance. In particular, if distributions $q$ and $q_{\text{approx}}$​ differ by at most $\delta$ in TV distance, then for any bounded measurable function $f \in [0,1]$, we can show the difference in conditional expectations over any event with non-negligible probability (at least $\gamma$) is bounded by:
> $$|E_{\mu \sim q}[f(\mu)|A] - E_{\mu \sim q_{approx}}[f(\mu)|A]|\leq \frac{2\delta}{\gamma}.$$
> Since $\gamma$ is purely from the agent's decision, it is unaffected by the noise level $\delta$. This allows us to quantify the change in FN and FP losses due to errors in $q$. In particular, both the FN and FP losses take the form of conditional expectations over events defined by thresholds (e.g.,$\mu_0 \leq \mu_b$​), multiplied by ratios of marginal probabilities. We can show that both terms change by at most $\frac{2\delta}{\gamma}$ if the total variation distance between $q$ and $q_{\mathrm{approx}}$​ is at most $\delta$, and the relevant conditioning events have probability at least $\gamma$ under both distributions. In short, we can give an approximation error bound of the form ($\text{Loss}$ can be either FP or FN): $|\text{Loss}(q) - \text{Loss}(q_{\text{approx}})| \leq \frac{4\delta}{\gamma}$. We will include the formal proof of this result and a discussion on the effect of noisy/imperfect observations in the revised version.
>
> > **Agent's imperfect observation of $\mu_0$**
>
> As for the agent (drug manufacturer), suppose that instead of exactly observing $\mu_0$, they have some unbiased estimator of it, $\hat{\mu}$. In extending our results to this unbiased estimate setting, the agent participation decision is no longer a threshold/deterministic; instead of observing $\mu_0$ and taking the participation decision $1[\mu_0 \geq \mu_{\tau}(\alpha)]$, now they observe $\hat{\mu}$ and their participation decision is given by $P[\hat{\mu} > \mu_{\tau}(\alpha)]$ – a softer decision boundary. While key properties like monotonicity in $\mu_0$ still hold if the estimator, the analysis needs to be re-done with this change, and will now include parameters relating to the distribution of the estimator (like variance).
>
> ---
>
> > **Interpreting our conclusion on cost and revenue dependent p-value threshold**
>
> Our results highlight that when agents strategically choose their trial samples and participation based on private beliefs and cost-revenue calculus, the p-value decision must also consider such dynamics. There are two ways to interpret this:
> 1. As pointed out in the review, one perspective is to set the p-value based on the expected revenue-cost calculus. This implies that for low-revenue high-development-cost drugs, the p-value threshold should be more generous; otherwise, it is not in the best interest of developers to participate. This can be interpreted as setting a domain/market-specific p-value.
> 2. In many cases, the first interpretation may not be palatable, and agencies may be reluctant to change p-values. In such cases, our model suggests that the principal can still derive optimal results against strategic agents by adjusting the other parameters. Specifically, we imagine regulators providing subsidies for development costs and trials to ensure optimal participation on the desired p-value. This is not far-fetched since governments often subsidize crucial drug developments (the COVID-19 vaccine, for example). From this perspective, our result can be interpreted as giving an accurate and interpretable optimal “cost subsidy” needed for drugs to pass under a given confidence interval and charge a given price.
>
> Thank you for bringing this up, and we think the latter interpretation is especially insightful. We aim to discuss this in the revised version.
>
> ---
>
> > **Additional work to convince regulatory bodies**
>
> Our initial results suggest that the current pricing/revenue of drugs in the US under the FDA p-value of 0.05 is consistent with our model of agent strategic behaviour. That said, from a regulatory perspective, more precise and fine-grained empirical work would be needed to further substantiate this.
>
> A more detailed discussion on the mitigation strategies, whether through adjusting p-values or subsidizing development costs (as mentioned above), is also warranted. Policy decisions like this naturally have broader externalities, hidden or explicit, and they need to be carefully assessed. As for the welfare of such strategies, it is possible to define a Price of Anarchy for Stackelberg games where the optimal welfare corresponds to maximizing the utility for all agents. Conceptually, however, this may be a bit odd since it is not clear why any objective beyond minimizing total error (the current principal objective being optimized) should be socially desirable. It may, however, be prescient to consider whether subsidizing development costs to encourage worthy participation is indeed optimal if there are budget constraints, or what the right trade-offs here are.

---

> > ### Author Response · Authors · 2025-08-07
> >
> > Dear Reviewer uvQJ, thank you again for your valuable feedback! As the Author–Reviewer discussion period comes to an end, please let us know if there are any remaining questions we can help clarify.

---

> > > ### Comment · Reviewer_uvQJ · 2025-08-08
> > > **Response to Rebuttal**
> > >
> > > Thank you for the thorough responses to my questions! I especially appreciate the refined interpretation #2, thinking of implications on subsidies is indeed very practical and relevant. I maintain my positive score that recommends acceptance.

---

### Official Review · Reviewer_s2e9 · 2025-07-15

**Clarity:** 3
**Significance:** 3
**Originality:** 3
**Rating:** 5
**Confidence:** 3

**Summary:**

This paper studies a very interesting model in which a principal and an agent are involved in hypothesis testing. The principal must decide whether to accept a new product (say a new drug) based on how effective it is compared to an existing baseline $\mu_b$. The agent knows how effective their sample is $\mu_0$ which is drawn from a distribution $q$. This is a Stackleberg game in which the principal first decides a threshold $\alpha$ for acceptance, which is decided by seeing some number of Bernoulli samples. The agent, upon knowing this, will decide whether to participate and, if so, choose and collect some number of samples $n \in [n_{\min}, n_{\max}]$. Each collection will cost $c_0 + cn$, where $c_0$ is a fixed cost and $c$ is a unit cost. If the product is accepted, the agent receives a reward of
$R - c - cn$. The main goal is to design the threshold $\alpha$ to minimize the loss due to false positives and false negatives (appropriately weighted).

In terms of techniques, the analysis is fairly intuitive but getting the details right takes care. The authors first show that for a given alpha, there is a particular threshold $\mu_T$ such that the agent will only participate if their value is above it. Furthermore it is computable efficiently and that as their value $\mu$ increases so does their utility. Now they show that they can bound the loss due to false positives and negatives for any given choice of alpha by breaking down the losses in each component.

Finally, they give a show case for drug companies and approvals for the FDA. They show that the their model roughly captures the economic dynamics of drug approvals in the United States. One interesting recomendation is that to decrease drug prices, the current threshold might be too low to incentivize low revenue but highly effective drugs.

**Questions:**

N/A

**Ethical Concerns:**

["NO or VERY MINOR ethics concerns only"]

**Quality:**

3

**Strengths And Weaknesses:**

The paper studies a very natural variant of the principal agent model that is highly relevant in FDA drug trials. They show that the game has a natural equilibrium that is computable for the principal and the agents to minimize the total rate of false positives and false negatives. The paper is well written and the technical sections are thorogh.

 The main weakness is that the model is natural and nice but the results are not entirely surprising. In fact, technically the paper is quite standard and doesn't innovate in any meaningful way.

 Nonetheless, I find the model useful and the practical application compelling. Perhaps EC would be a better conference for this work but I recommend acceptance.

---

> ### Author Rebuttal · Authors · 2025-07-30
>
> Thank you for your helpful review and feedback! We present a natural, tractable model, grounded in real‑world concerns, for studying hypothesis testing in the presence of strategic participants. Our analysis builds on a sequence of subtle observations to produce clear, interpretable results that give valuable insights on incentives and behaviours in this setting.
>
> Depending on the perspective, the resulting conclusions can be surprising. The fact that a lower bound on the optimal p-value emerges from our work is distinct from the classic thinking of upper-bounding the p-value to control Type I errors. Our analysis is also flexible, highlighting that if context dependent p-values are unpalatable, regulators can instead subsidize the fixed and marginal costs to achieve similar results. We will highlight the latter point more in the revision.
>
> We view our work as part of the broader research direction on data-driven decision-making under strategic behavior. Strategic classification, strategic regression, and incentive-aware machine learning have received significant attention in the past decade, with many contributions appearing in NeurIPS and other leading machine learning venues. We hope our work encourages further exploration into strategic aspects of statistical analysis, an area that remains relatively under-explored.

---

> > ### Comment · Reviewer_s2e9 · 2025-08-08
> >
> > Thank you for the response. I am happy with the response and will keep my score the same.

---

### Decision · Program_Chairs · 2025-09-17

**Decision:**

Accept (spotlight)

**Comment:**

**Summary**

The paper studies problems where a principal must decide between approving or rejecting a product manufactured by an agent based on evidence provided by the agent. The paper tackles the problem by using game-theoretic tools in contract theory, by providing an interesting characterization of the principal’s optimal p-value. The paper also empirically validates the model and results by using publicly available data on drug approvals.

**Strengths**

The paper is very well written, and the results presented are of broad interest. The model is a natural variant of the classical principal-agent problem that may be relevant in many application scenarios.

**Weaknesses**

The results are *not* entirely surprising. However, the model is natural and worth studying.

**Decision**

All the Reviewers agree that this is a good paper for NurIPS, and thus I recommend its **acceptance**.